

# The number fraction of iron-containing particles affects OH, HO$_2$ and H$_2$O$_2$ budgets in the atmospheric aqueous phase

Amina Khaled[1], Minghui Zhang[1], and Barbara Ervens[1]

[1]Université Clermont Auvergne, CNRS, SIGMA Clermont, Institut de Chimie de Clermont-Ferrand, 63000 Clermont-Ferrand, France

**Correspondence:** Barbara Ervens (barbara.ervens@uca.fr)

**Abstract.** Reactive oxygen species (ROS), such as OH, HO$_2$ and H$_2$O$_2$) affect the oxidation capacity of the atmosphere and cause adverse health effects of particulate matter. The role of transition metal ions (TMIs) in impacting the ROS concentrations and conversions in the atmospheric aqueous phase has been recognized for a long time. Model studies usually assume that the total TMI concentration as measured in bulk aerosol or cloud water samples is distributed equally across all particles or droplets. This assumption is contrary to single-particle measurements that have shown that only a small number fraction of particles contain iron and other TMIs ($F_{N,Fe} < 100\%$) which implies that also not all cloud droplets contain TMIs. In the current study, we apply a box model with an explicit multiphase chemical mechanism to simulate ROS formation and cycling in (i) aqueous aerosol particles and (ii) cloud droplets. Model simulations are performed for the range of $1\% \leq F_{N,Fe} \leq 100\%$ for constant pH values of 3, 4.5 and 6 and constant total iron concentration (10 or 50 ng m$_{air}^{-3}$). Model results are compared for two sets of simulations with $F_{N,Fe} < 100\%$ (FeN<100) and 100% (FeBulk). We find largest differences between model results in OH and HO$_2$/O$_2^-$ concentrations at pH = 6. Under these conditions, HO$_2$ is subsaturated in the aqueous phase because of its high effective Henry's law constant and the fast chemical loss reactions of the O$_2^-$ radical anion. As the main reduction of process of Fe(III) is its reaction with HO$_2$/O$_2^-$, we show that the HO$_2$ subsaturation leads to predicted Fe(II)/Fe(total) ratios for $F_{N,Fe} < 100\%$ that are lower by a factor of $\leq 2$ as compared to bulk model approaches. This trend is largely independent of the total iron concentration, as both chemical source and sink rates of HO$_2$/O$_2^-$ scale with the iron concentration. The chemical radical (OH, HO$_2$) loss in particles is usually compensated by its uptake from the gas phase. We compare model-derived reactive uptake parameters $\gamma$(OH) and $\gamma$(HO$_2$) for the full range of $F_{N,Fe}$. While $\gamma$(OH) is not affected by the iron distribution, the calculated $\gamma$(HO$_2$) range from 0.0004 to 0.03 for $F_{N,Fe}$ = 1% and 100%, respectively. Implications of these findings are discussed for the application of lab-derived $\gamma$(HO$_2$) in models to present reactive HO$_2$ uptake on aerosols. As the oxidant budget in aerosol particles and cloud droplets is related to the oxidative potential, we also conclude that the iron distribution ($F_{N,Fe}$ should be taken into account to estimate the ROS concentrations and health impacts of particulate matter that might be overestimated by bulk sampling and model approaches. Our study suggests that the number concentration of iron-containing particles may be more important than the total iron mass concentration in determining ROS budgets and uptake rates in cloud and aerosol water.



## 1 Introduction

The main oxidants in the atmospheric aqueous phases of cloud and aerosol particles include the hydroxyl radical (OH) and hydrogen peroxide ($H_2O_2$) whose concentration levels are closely linked to the hydroperoxy radical ($HO_2/O_2^-$). The concentrations of these reactive oxygen species (ROS) are influenced by various redox reactions of transition metal ions (TMIs). Iron is the most abundant TMI in aerosol particles and cloud water. Its main sources include dust and coal combustion (Moffet et al., 2012) and biomass burning (Wang et al., 2015). Its concentration in cloud water ranges from nano- to micromolar levels (Cini et al., 2002; Deguillaume et al., 2005). Iron mass concentrations in aerosol samples reach from less than 1 ng m$^{-3}$ to more than 100 ng m$^{-3}$ (Schmücke, Germany) and sometimes up to several hundred ng m$^{-3}$ at other continental locations (Fomba et al. (2015) and references therein).

The role of transition metal ions in affecting the ROS concentration levels and redox reactions in clouds and particles has been explored in many model, lab and field studies, e.g., Deguillaume et al. (2004, 2005); Mao et al. (2013). Reactions of iron and other TMIs (e.g., copper) in the atmosphere have also been linked to the oxidative potential of particulate matter, causing oxidative stress in the respiratory tract and lungs (Saffari et al., 2014; Arangio et al., 2016; Tong et al., 2016; Molina et al., 2020; Wei et al., 2021). In particular, the Fenton reaction, the oxidation of iron(II) by hydrogen peroxide has been identified as one of the main chemical sources of the OH radical in cloud water (Ervens et al., 2003; Deguillaume et al., 2004; Tilgner et al., 2013), aqueous phase aerosol particles (Al-Abadleh, 2015) and lung fluid (Charrier et al., 2015).

TMI concentrations in the cloud and aerosol phases are usually reported based on bulk measurements representing the total TMI mass per aqueous or gas phase volumes. However, single-particle analyses have shown that iron is only present in a small number fraction of particles. Furutani et al. (2011) found that up to 15% of particles in the diameter range between 0.5 and 1 $\mu$m contain iron (Zhang et al., 2014). Similar number fractions (< 1 – 15%) were observed in Shanghai, where air masses were also affected by dust storms or biomass and coal burning. A similar size range for iron-containing particles but smaller average number fraction ($\sim$4%) were found above the English Channel in air masses affected by steel works (Choël et al., 2007). As particles in this size range commonly act as cloud condensation nuclei (CCN), these analyses suggest that not all cloud droplets contain iron and that also the measured iron mass concentration in aerosol populations is not equally distributed among all particles. Drop-size-resolved cloud water measurements at a continental background site have shown that iron and copper are present in the same drop size range whereas manganese is more abundant in larger droplets (Fomba et al., 2015). This may suggest that CCN were comprised of internal mixtures of iron and copper whereas manganese-containing particles were externally mixed and of different hygroscopicity and/or different sizes.

The oxidation state of iron affect its solubility and thus its bioavailability and biogeochemical cycles in the atmosphere and oceans. Generally, ferrous salts are more soluble than ferric salts, with increasing solubility under acidic conditions, e.g. Ingall et al. (2018). Results from a global aerosol model study revealed large differences in predicted and observed iron solubility; observed Fe(II)/Fe(total) ratios were on average between $\sim$1 and $\sim$10% with larger variability in coarse than in fine particles





(Luo et al., 2005). The Fe(II) fractions in particle samples above oceans has been shown to be above 30% - 75% (Ingall et al., 2018). Measurement of the Fe(II)/Fe(total) ratio on a single particle basis revealed ratios of $\leq 20\%$ for particles from various sources (Takahama et al., 2008). Fe(II) often dominates the total iron in cloud water, in particular during day-time when

iron(III) hydroxy and organo complexes are reduced by photolysis processes (Deguillaume et al., 2005).

Multiphase chemistry models are usually initialized with bulk concentrations of iron to make predictions on the role of aqueous phase chemistry on ROS levels (e.g., Ervens et al. (2003); Tilgner et al. (2013); Tong et al. (2017)), sulfate formation (e.g., Alexander et al. (2009); Chang et al. (1987)) or iron deposition (Myriokefalitakis et al., 2018). In a previous model study, we have demonstrated that such bulk approaches may not be appropriate for highly reactive compounds as redistribution and

diffusion processes among droplets of chemical composition might lead to non-linear effects impacting concentration levels (Khaled et al. (2021)). While that study was focused on the biodegradation of organics by bacteria that are only present in a small number fraction of cloud droplets, we apply the same idea in the present study to the ROS cycling dependent on the number fraction of iron-containing particles.

We perform box model simulations a box model with a detailed gas and aqueous phase chemical mechanism, in which a

constant iron mass concentration [ng m$_{air}^{-3}$] is distributed to a fraction of (i) cloud droplets or (ii) aqueous aerosol particles. We investigate the conditions in terms of pH, iron distribution and total iron mass, under which the number fraction of iron-containing particles significantly affects the concentration levels and phase transfer of OH, HO$_2$ and H$_2$O$_2$ in the atmospheric multiphase system. We discuss the potential implications of these effects for the interpretation of measurements and model studies of the iron oxidation state, radical uptake rates onto aqueous aerosol particles and ROS budgets and oxidative potentials

of aerosol and cloud water.

## 2    Multiphase box model

### 2.1    Model description

We use a box model with detailed gas and aqueous phase chemical mechanisms. The gas-phase chemical mechanism with 58 reactions is based on the NCAR Master Chemical Mechanism (Madronich and Calvert, 1989). The aqueous phase chemical

mechanism includes 43 reactions (Ervens et al., 2008; Tong et al., 2017) (Tables S1 - S3 in the supplement). Phase transfer processes for 14 species are described kinetically based on the resistance model (Schwartz, 1986; Nathanson et al., 1996) (Table S4). The model uses the standard equations for multiphase processes:

$$\frac{dC_{aq}}{dt} = \underbrace{k_{mt} \, LWC \left( C_g - \frac{C_{aq}}{LWC \, K_{H(eff)} \, R \, T} \right)}_{phase \; transfer \; rate} + \underbrace{S_{aq} - L_{aq}}_{chem \; rate} \tag{E.1}$$

$$\frac{dC_g}{dt} = \underbrace{-k_{mt} \, LWC \left( C_g - \frac{C_{aq}}{LWC \, K_{H(eff)} \, R \, T} \right)}_{phase \; transfer \; rate} + \underbrace{S_g - L_g}_{chem \; rate} \tag{E.2}$$






whereas both the gas phase ($C_g$) and aqueous phase ($C_{aq}$) concentrations are expressed in units of mol $g_{air}^{-1}$, *LWC* is the liquid water content [vol/vol], $K_{H(eff)}$ is the (effective) Henry's law constant, and $S_{aq}$, $S_g$ and $L_{aq}$, $L_g$ are the chemical source and loss rates in the aqueous and gas phases [mol $g_{air}^{-1} s^{-1}$]. *R* and *T* are the constant for ideal gases (0.082058 L atm (K mol)$^{-1}$) and absolute temperature [K]. The mass transfer coefficient $k_{mt}$ [s$^{-1}$] is defined as

$$k_{mt} = \left( \frac{r_d{}^2}{3\,D_g} + \frac{r_d}{3\,\alpha} \sqrt{\frac{2\pi\,M_g}{R\,T}} \right)^{-1} \tag{E.3}$$

whereas $r_d$ is the drop radius, $D_g$ the gas phase diffusion coefficient, $\alpha$ the mass accommodation coefficient, and $M_g$ the molecular weight. The box model considers monodisperse cloud droplet or wet aerosol particle populations with constant diameters (*D*) of 20 $\mu$m for cloud droplets and 300 nm for aqueous aerosol particles, respectively.

## 2.2 Model simulations


The schematic of the model setup together with the main ROS formation and conversion processes are shown in Fig. 1. Results of two model approaches will be compared:

- a) **FeBulk**: All droplets (particles) have the same chemical composition. The iron concentration is identical in all droplets (particles). The fraction of iron-containing droplets (particles) is $F_{N,Fe} = 100\%$.

- b) **FeN<100**: A subset of the number concentration (N) of droplets (or particles) is initialized with iron whereas the remainder of the droplets (particles) is iron-free. In the base case simulations, we assume that the number fraction of iron-containing droplets (particles) is $F_{N,Fe}= 2\%$; $F_{N,Fe}$ is varied from 1 - 99% in sensitivity studies (Section 3.3).

For most simulations, we assume an iron mass concentration ($m_{Fe}$ = 10 ng $m_{air}^{-3}$) (Sensitivity studies for $m_{Fe}$ = 50 ng $m_{air}^{-3}$ are discussed in Section 3.3.1.) Therefore, in the FeN<100 approach, the aqueous phase iron concentration [M] is higher in the
iron-containing droplets (particles) than in the FeBulk approach as the iron mass is distributed to fewer droplets (particles). All model parameters are summarized in Figure 1. Simulations are performed for constant values of pH = 3, 4.5 and 6. All model parameters are summarized in Table 1.

Chemical and phase transfer are analysed after a simulation time of 400 seconds. Gas phase concentrations are not replenished throughout the simulations; their initial concentrations are listed in Table S5. Therefore, initial trace gases (e.g. SO$_2$)
may be efficiently consumed, affecting the overall chemical rates and their relative contributions on longer time scales (Section 3.2).





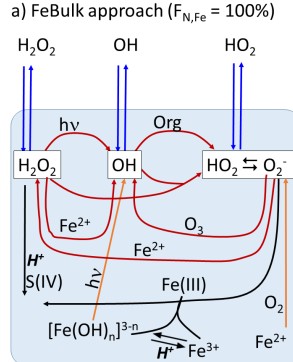
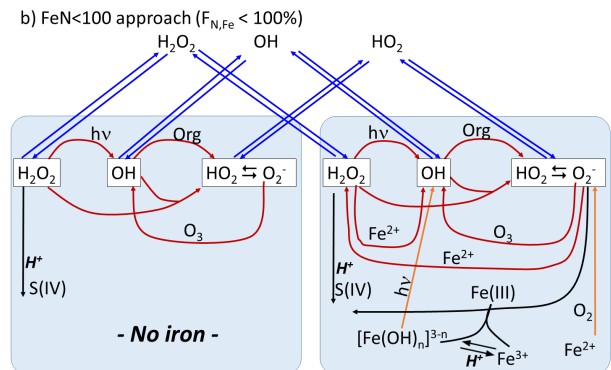

**Figure 1.** Model schematic: a) FeBulk: The total iron mass concentration (base case: $m_{Fe}$ = 10 ng m$^{-3}$) is equally distributed among all droplets or particles ($F_{N,Fe}$ = 100%), b) FeN<100: The same iron mass concentration is distributed among a subset of droplets or particles (base case: $F_{N,Fe}$ = 2%). Major ROS processes are phase transfer (blue), ROS net production (orange), ROS conversions (red) and ROS loss (black). Model parameters are summarized in Table 1.

## 3   Model results and discussion

### 3.1   Aqueous phase concentrations and rates

#### 3.1.1   Cloud droplets

Figure 2 summarizes the ROS concentrations, and the chemical and phase transfer rates for the cloud case ($m_{Fe}$ = 10 ng m$^{-3}$, $F_{N,Fe}$ = 100% vs 2%, pH = 4.5, t = 400 s). The corresponding figures for simulations at pH = 3 and 6 are shown in Fig. S1 (Supplement). The net rates of the phase transfer [mol $g_{air}^{-1} s^{-1}$] into the aqueous phase are shown in the yellow boxes. For FeN<100, the relative contributions [%] of this rate into the iron-free and iron-containing droplets are shown next to the arrows. If the effect of the iron distribution across the droplet population were negligible, the ratio of these rates should be equal to the

ratio of the LWCs of the two drop classes (98% : 2%). The phase transfer rates are additionally shown in units of M s$^{-1}$, (using the air density of 1.13·10$^{-3}$ g cm$^{-3}$ and LWCs in Table 1), together with the chemical source and loss rates in the aqueous phase [M s$^{-1}$] and the corresponding aqueous phase ROS concentrations [M] (white boxes). If the effect of iron were negligible on aqueous phase concentrations and aqueous phase rates, there should be no difference in these values in any of the droplets, neither between the results for FeBulk and FeN<100 nor between the iron-free and iron-containing droplets.

All ROS are predicted to be taken up into cloud water, independently of the iron distribution at pH = 4.5 (Fig. 1). For all three ROS, the ratio of the phase transfer rates near the ratio of the LWCs of the two droplet classes (98% : 2%). Significant deviations from his theoretical value occur at pH = 3 for HO$_2$ (Fig. S1b) and at pH = 6 for H$_2$O$_2$ that is predicted to evaporate from iron-free droplets (Fig. S1d).



**Table 1.** Model parameters for multiphase simulations for cloud droplet and aerosol populations. $D$ = diameter, $N$ = number concentration, $LWC$ = liquid water content, $m_{Fe}$ = total iron mass concentration, $F_{N,Fe}$ = number concentration of droplets or particles that contain iron, $[Fe]_{aq}$ = aqueous phase concentration of iron. Numbers in parentheses denote values or parameter ranges of sensitivity studies; all other values are those of base case simulations.

| Parameter | FeBulk | FeN<100 | |
|---|---|---|---|
| Cloud droplets | All | Iron-free | Iron-containing |
| $D$ / $\mu$m | 20 (10, 40) | 20 (10, 40) | 20 (10, 40) |
| $N$ / cm$^{-3}$ | 100 | 98 (99 - 1) | 2 (1 - 99) |
| $LWC$ / g m$^{-3}$ | 0.42 | 0.41 (0.416 - 0.042) | 2 (0.042 - 0.416) |
| $m_{Fe}$ / ng m$^{-3}$ | 10 (50) | 0 | 10 (50) |
| $F_{N,Fe}$ / % | 100 | 0 | 2 (1 - 99) |
| $[Fe]_{aq}$ / $\mu$M | 0.41 | 0 | 21 (0.43 - 42) |
| Aerosol particles | All | Iron-free | Iron-containing |
| $D$ / $\mu$m | 0.3 (0.15, 0.6) | 0.3 (0.15, 0.6) | 0.3 (0.15, 0.6) |
| $N$ / cm$^{-3}$ | 1500 | 1470 (1485 - 15) | 2 (15 - 1485) |
| $LWC$ / $\mu$g m$^{-3}$ | 21 | 20.6 (20.8 - 0.3) | 0.6 (0.3 - 20.8) |
| $m_{Fe}$ / ng m$^{-3}$ | 10 (50) | 0 | 10 (50) |
| $F_{N,Fe}$ / % | 100 | 0 | 2 (1 - 99) |
| $[Fe]_{aq}$ / M | 0.008 | 0 | 0.41 (0.00008 - 0.82) |

The uptake rates in M s$^{-1}$ between FeBulk and the iron-free droplets in the FeN<100 approach are similar (e.g. for $H_2O_2$: 0.56 M s$^{-1}$ and 0.53 M s$^{-1}$) whereas they are higher into the iron-containing droplets (2.2 M s$^{-1}$).

The $H_2O_2$ concentrations in all droplets for all conditions and pH values (Fig. S1) correspond to the equilibrium concentrations according to Henry's law, in agreement with previous studies that have shown that $H_2O_2$ is at thermodynamic equilibrium in cloud water (Ervens, 2015). Accordingly, the partitioning coefficient $\epsilon$ is near unity for FeBulk and FeN<100 (Fig. 3a). It is defined as

$$\epsilon = \frac{C_{aq}}{p_g \cdot K_{H(eff)}} \tag{E.4}$$

with $C_{aq}$ and $p_g$ being the aqueous phase concentration [M] and gas phase partial pressure [atm] and $K_{H(eff)}$ the (effective) Henry's law constant [M atm$^{-1}$].

At pH = 3 and 4.5, the efficient $H_2O_2$ consumption is compensated by a relatively higher uptake rate, resulting in the efficient replenishment of $H_2O_2$ and a constant (equilibrium) concentration. Since these droplets only comprise 2% of the total LWC, this difference is not reflected in the net uptake rate. At pH = 6 (Fig. S1c and d), the majority of cloud droplets are predicted to be a source of $H_2O_2$ as the chemical loss rate in the FeBulk droplets and in the iron-free droplets (98% of total) drop by a factor of ∼10 between pH = 3 and 6 whereas the source rate slightly increases. This results in a slight supersaturation of $H_2O_2$ in the





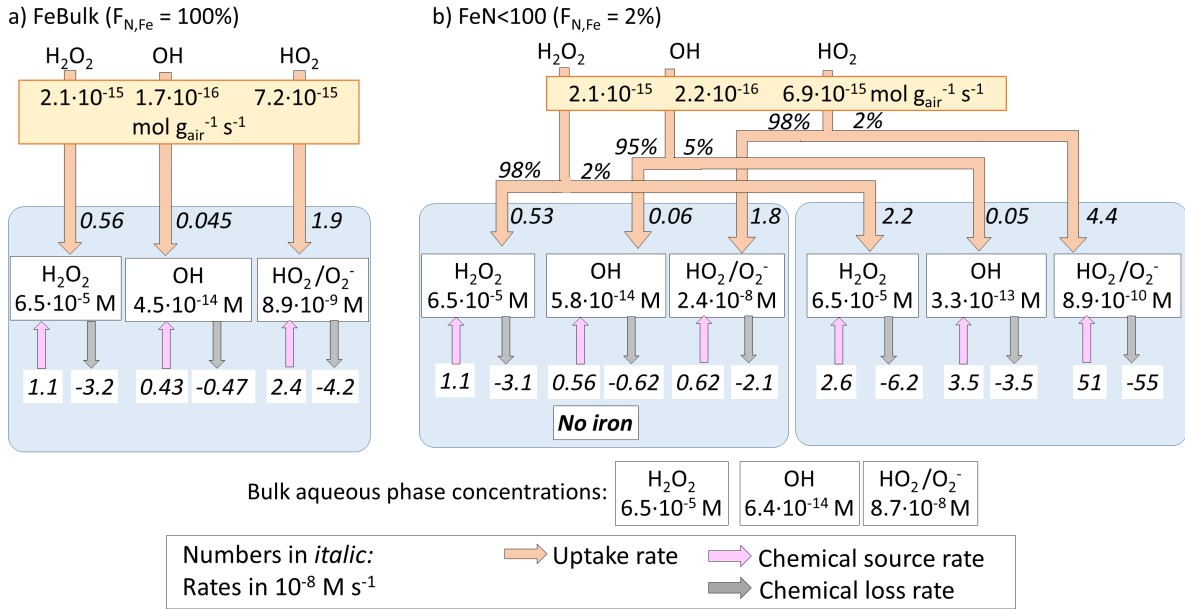

**Figure 2.** Multiphase scheme for cloud water showing the phase transfer and chemical source and loss rates in the aqueous phase, $m_{Fe}$ = 10 ng m$^{-3}$, pH = 4.5 a) FeBulk approach, b) FeN<100 approach with a fraction of iron-containing droplets of $F_{N,Fe}$ = 2%. Numbers below the species names are aqueous phase concentrations [M], chemical and phase transfer rates are shown in M s$^{-1}$. Net phase transfer rates near the top of the figure are expressed in gas phase units [mol g$_{air}^{-1}$ s$^{-1}$]; the contributions [%] into the iron-free and iron-containing droplets are shown next to the arrows. The bulk concentrations below panel b) represent the total ROS concentrations weighted by the contributions of the two droplet classes to total LWC (98% : 2%).

aqueous phase ($\epsilon \sim 1.05$). The reasons for the pH dependence of the ROS concentrations are discussed in detail in (Section 3.2). The OH concentrations differ by nearly one order of magnitude between the iron-free and iron-containing cloud droplets

(e.g., at pH = 4.5: 5.8·10$^{-14}$ M vs 3.3·10$^{-13}$ M, Fig. 2b); the concentration in the FeBulk model is even slightly lower (4.5·10$^{-14}$ M, Fig. 2a). This difference in OH concentrations between the two droplet classes is also reflected in the higher partitioning coefficient $\epsilon$(OH) for the iron-containing droplets as compared to the iron-free droplets and in the FeBulk simulations (Fig. 3b). This trend implies that the gas phase OH concentrations are not significantly different for FeBulk or FeN<100.

The total OH concentration in the cloud water is calculated as a weighted average of the LWCs comprising the iron-free and

iron-containing droplets. These concentrations are shown as bulk aqueous phase concentration at the bottom of Fig. 2b and Fig. S1a,b and c,d, respectively. These concentrations corresponded to those in bulk cloud water samples from cloud droplet populations with $F_{N,Fe}$ = 2% (provided that chemical processes in the sample upon sampling do not change the ROS levels). The total OH concentration is approximately 50% higher (4.5·10$^{-14}$ M for FeBulk as compared to 6.4·10$^{-14}$ M for FeN<100, at pH = 4.5. A much greater discrepancy of nearly one order of magnitude between the two approaches is predicted at pH =





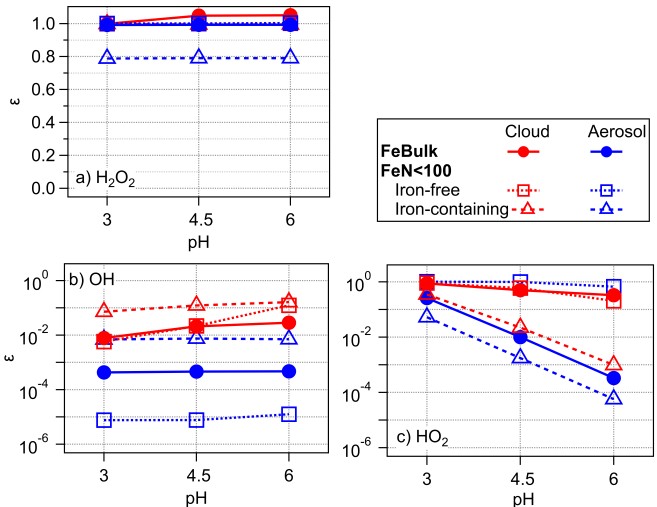

**Figure 3.** Partitioning coefficients $\epsilon$ for a) $H_2O_2$, b) OH and c) $HO_2$ for pH=3 , 4.5 and 6 for FeBulk (filled symbols) and FeN<100 in iron-free (squares) and iron-containing droplets (triangles). Results are shown for simulations of cloud droplets (red) and aerosol particles (blue) at three pH values (3, 4.5, and 6); lines are added to guide the eye.

6, whereas the difference in the bulk concentrations at pH = 3 is smaller. For all cases, the chemical source and loss rates of the OH radical compensate each other, leading to a relatively small uptake rate of the moderately soluble OH radical into the aqueous phase ($\sim 10^{-16}$ mol g$^{-1}$ s$^{-1}$) and not significantly affected by the iron distribution ($F_{N,Fe}$).

The concentration difference of $HO_2/O_2^-$ between iron-free and iron-containing droplets at pH = 4.5 is nearly two orders of magnitude ($2.4 \cdot 10^{-8}$ M vs $8.9 \cdot 10^{-10}$ M) whereas the concentration for FeBulk are predicted in between these two values ($8.9 \cdot 10^{-9}$ M). A similar trend is also seen for the $\epsilon$ values is between the values for the two drop classes for FeN<100 (Figs. 4b and 3c). The difference in the concentrations is even greater (> factor 200) at pH = 6 (Figure S1d) whereas they are less than a factor of 3 at pH = 3. These trends suggest that the concentrations are dependent on pH (Section 3.2). The comparison of the bulk aqueous phase $HO_2/O_2^-$ concentrations to those in FeBulk shows that at pH = 4.5 and 6, the total $HO_2/O_2^-$ is underestimated by about one order of magnitude if $F_{N,Fe} = 2\%$. The absolute differences between the chemical source and loss rates in the iron-containing droplets is largest ($66 \cdot 10^{-8}$ M s$^{-1}$ vs $74 \cdot 10^{-8}$ M s$^{-1}$) at pH = 3, even though the ratio of these rates is approximately the same at all pH values. This imbalance in the chemical rates leads to a very efficient $HO_2$ uptake into iron-containing droplets. Consequently, the phase transfer into these droplets comprises 13% of the total uptake rate (Fig. S1b) whereas at the higher pH values the contributions correspond to the ratio of LWCs between the two droplets classes (98% : 2%). However, since the effective Henry's law constant of $HO_2$ is relatively small at low pH, $HO_2$ is nearly at thermodynamic equilibrium ($\epsilon \sim 1$, Fig. 3c). The higher Henry's law constant of $HO_2$ and rate constants of $O_2^-$ reactions as compared to those of $HO_2$ at higher pH leads to an increasing subsaturation, i.e. to $\epsilon \sim 10^{-3}$ in iron-containing droplets whereas the partitioning of $HO_2$ in the iron-free droplets and in the FeBulk approach correspond nearly to their equilibrium value (Fig. 3c).





### 3.1.2 Aqueous aerosol particles

Since the LWC of the aqueous aerosol particles is smaller by a factor of $2 \cdot 10^5$ than in clouds, the aqueous phase iron concen-

175 tration is accordingly higher and so are all chemical rates of iron reactions. Similar to cloud water, the $H_2O_2$ concentration is neither significantly affected by the iron distribution nor by iron reactions, i.e., at a given pH value its aqueous phase concentration is nearly the same in all particles. At pH = 4.5 and 6, evaporation of $H_2O_2$ is predicted for FeBulk and from the iron-free particles in FeN<100 (Fig. S2c and d). However, since the uptake of $H_2O_2$ into the iron-containing droplets is much more efficient, exceeding the evaporation rate by a factor of 25 (-4% vs 104%, Fig. 4b) and ~2 (-143% vs 243% at pH = 6,

180 Figure S2d), aqueous particles represent a net sink of $H_2O_2$.

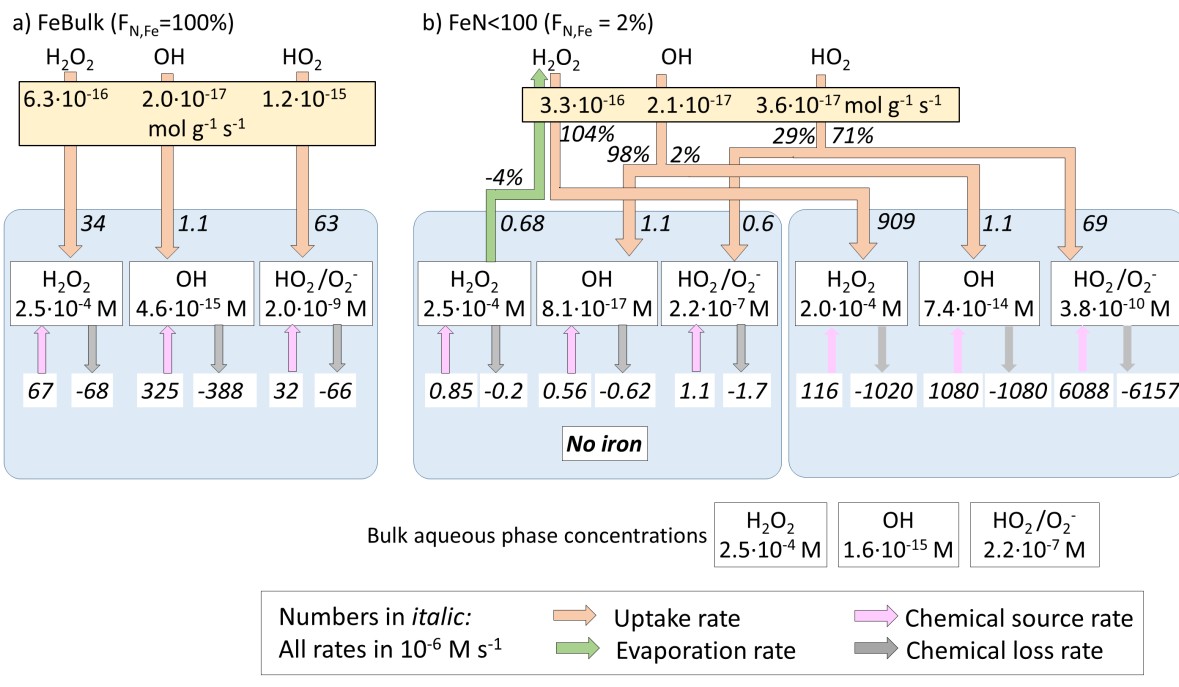

**Figure 4.** Same as Fig. 2 but for aerosol case (model parameters of the base case in Table 1)

.

Similar to the cloud case, the OH concentrations in the aqueous phase of FeBulk are between those in the iron-free and iron-containing particles for FeN<100. The difference between the concentration in iron-free and iron-containing particles is up to three orders of magnitude (pH = 3, Figure S2b), which is also reflected in the same difference in $\epsilon$ (Fig. 3b). The OH concentrations do not show a strong pH dependence but their bulk values are about a factor of three lower for $F_{N,Fe} = 2\%$ than

185 for $F_{N,Fe} = 100\%$ ($1.6 \cdot 10^{-15}$ M vs ~$4.6 \cdot 10^{-15}$ M). Even at the high iron concentrations as present in aerosol water ($\leq 0.41$ M), the chemical source and loss rates of OH nearly cancel, leading to a relatively small uptake rate.





The $HO_2/O_2^-$ in the FeBulk approach are comparable to those as predicted for cloud water. However, the concentration differences between the iron-free and -containing particles are four orders of magnitude at pH = 6 ($1.5 \cdot 10^{-6}$ M and $1.3 \cdot 10^{-10}$ M, Figure S2d). The resulting bulk aqueous phase concentrations are at least one order of magnitude underestimated by FeBulk (pH = 3); this difference is even a factor of 2000 at pH = 6 (Figs. 4d and S2d). While the chemical loss rate of $HO_2/O_2^-$ is nearly equal or even larger than the source rate at pH = 4.5 and 6, respectively, at pH = 3 the source rate exceeds the loss rate resulting in evaporation of $HO_2$ (Fig. S2b). However, since the uptake rate into the iron-containing particles is larger than this evaporation rate, the net flux of $HO_2$ is an uptake into the particle phase.

## 3.2 Relative contributions of chemical reactions and phase transfer to ROS source and loss rates in the aqueous phase

Generally, a high uptake rate reflects efficient chemical loss and/or inefficient chemical production in the aqueous phase. To understand these trends of concentrations and rates as a function of pH as discussed in the previous section, Fig. 5 summarizes the main chemical pathways for the three ROS for the cloud and aerosol cases for the same conditions as in Figs. 2 and 4. The individual chemical source (S) and loss (L) processes are indicated between the figure panels for the cloud (top) and aerosol (bottom) cases. Only chemical processes are listed that contribute to more than 1% to the corresponding rate for any simulation. As the phase transfer (PT) can be source (i.e., uptake) or loss (i.e., evaporation) for the aqueous phase species, its contribution is placed between the chemical source and loss contribution. The relative contributions are related to the total source and loss rates [mol g$^{-1}$ s$^{-1}$] as indicated in the boxes in each panel.

Given its high Henry's law constant ($K_{H,H2O2}$ = $1.02 \cdot 10^5$ M atm$^{-1}$), the $H_2O_2$ uptake contributes nearly 50% to the total $H_2O_2$(aq) source at pH $\leq$ 4.5 in cloud water (Fig. 5a) and even more in aerosol water (50 - 80%) (Fig. 5d). However, as already shown in Figs. 2 and 4, the uptake occurs mostly (98%) into iron-free cloud droplets, whereas in the aerosol case, most $H_2O_2$ is taken up by the iron-containing particles. The main loss of $H_2O_2$ in cloud water is the pH-dependent reaction with sulfur(IV) (L1 in Fig. 5a) whose rate decreases with increasing pH (R4 in Table S1). Therefore, the chemical loss of $H_2O_2$ is very inefficient at pH = 6 in cloud water ($2.4 \cdot 10^{-15}$ mol g$^{-1}$ s$^{-1}$ at pH = 6 vs $\sim 11 \cdot 10^{-15}$ mol g$^{-1}$ s$^{-1}$ at pH $\leq$ 4.5, Fig. 5a). The efficiency of $H_2O_2$ by the recombination of $HO_2$ (S1 in Fig. 5a) increases with pH as the reaction of $HO_2$ and $O_2^-$ is much faster than with undissociated $HO_2$ ($100 \cdot k_{R5} \sim k_{R6}$, Table S1). Because of this increasing chemical source and decreasing loss rates with increasing pH, $H_2O_2$ evaporates from the cloud droplets at pH = 6. Figure 5 only represents a snapshot of the rates after a simulation time of 400 seconds. This time corresponds approximately to the lifetime of a single cloud droplet but underestimates the time an aerosol particle might be exposed to a given relative humidity. We have chosen this relatively short time, as in our box model setup, the initialized gases are not replenished over time as no emissions are considered. At longer time scales (after 2000 seconds), the conclusions would not be drastically different (Figure S5); the only major difference is the loss of $H_2O_2$ by sulfur(IV) (L1) as $SO_2$ is consumed quickly within the first few minutes.

In aerosol water, the Fenton reaction (L2) is the main loss reaction of $H_2O_2$. The main $H_2O_2$ source process (S2) is the reaction of $Fe^{2+}$ with $O_2^-$ (S2) whose rate increases with increasing pH value. Since in iron-free particles, the chemical loss rate (L1) decreases with pH, but its source rate (S1) increases, $H_2O_2$ evaporates from iron-free particles. In the iron-containing particles, both main source and sink reactions are directly dependent on the $Fe^{2+}$ concentration (S2, L2). As the iron con-





**Figure 5.** Relative contributions of chemical and phase transfer rates to the total sources and losses of the three ROS. The total rates [mol $g_{air}^{-1}$ $s^{-1}$] are shown in boxes of each panel; for the FeN<100 approach, the contribution in iron-free and iron-containing droplets (particles) are displayed as open and dashed bars. Simulations were performed at constant pH values of pH = 3 (red), 4.5 (black). and 6 (blue) for cloud conditions: a) $H_2O_2$, b) OH, c) $HO_2$ and aerosol conditions: d) $H_2O_2$, e) OH, f) $HO_2$. The chemical source (S) and loss (L) reactions in the aqueous phase are listed between the panels.





centration in the iron-containing particles is ∼50 times higher for FeN<100 than for FeBulk, it seems surprising at first that the chemical rates do not scale by this factor. The $HO_2/O_2^-$ concentration is much higher in iron-free than in iron-containing particles, both in the FeBulk and FeN<100 approach, in particular at pH = 6. Also this trend, thus, cannot explain the lower chemical $H_2O_2$ source rates of S2. Therefore, only a difference in the $Fe^{2+}$ concentration can explain the differences in the

chemical rates of the chemical $H_2O_2$ sources and sinks. Indeed, Fe(II) contributes only ∼20% to the total iron for $F_{N,Fe}$ = 2% whereas its contribution is ∼55% for the FeBulk approach ($F_{N,Fe}$ = 100%) (Fig. 6).

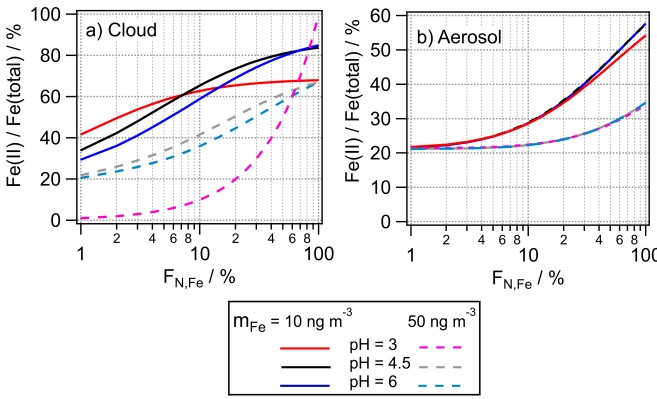

**Figure 6.** Predicted Fe(II)/Fe(total) ratios for the a) cloud and b) aerosol cases for pH=3 , 4.5 and 6. Solid lines are for $m_{Fe}$ = 10 ng m$^{-3}$; dashed lines are for $m_{Fe}$ = 50 ng m$^{-3}$

    Independently of the iron distribution, the relative importance of the uptake of the OH radical into cloud water decreases with increasing pH, from ∼30% at pH = 3 to < 10% at pH = 6 (Fig. 5b). As the rate of the $O_2^-$ reaction with ozone (S1) increases with pH, the relative importance of this reaction increases with increasing pH value. The loss of OH in the aqueous

phase are reactions with organic compounds; the rate of L1 in Fig. 5b and e represents the sum of all reaction rates of OH with organics in Table S1. As the rate constants of OH reactions with carboxylates are generally higher than those with their corresponding acids, the chemical loss rate in cloud water increases with pH. The most striking difference in the source and loss patterns of OH between cloud and aerosol water is the dominating role of the Fenton reaction (S2) as OH source in aerosol water. It contributes nearly 100% of the sources at all pH values, making the OH uptake as source negligible. Similar to the

trends as described above for $H_2O_2$, the chemical rates of the $Fe^{2+}$-dependent OH source (Fenton reaction, S2) is much smaller in for FeN<100 than for FeBulk, which can again be explained by the lower Fe(II) concentration at low $F_{N,Fe}$. As the uptake of OH is fairly inefficient due to its low Henry's low constant ($K_{H,OH}$ = 25 M atm$^{-1}$, Table S4), the chemical source rate is nearly fully compensated by the chemical loss rate that under all pH and LWC conditions resulting in (nearly) the same value as the source rate.

The only major sources of $HO_2/O_2^-$ are the reactions of $Fe^{2+}$ with molecular oxygen (S1) and the OH reactions with organics (S2). The slight increase in the overall reaction rate of S2 with pH explains the highest chemical source rate in all droplets at

highest pH. Its main loss reactions are the reactions of Fe(III) (both free $Fe^{3+}$ and the hydroxy complexes $[Fe(OH)_n]^{3n-1}$ with n = 1, 2). The dominance of these processes for the loss of $HO_2/O_2^-$ leads to very efficient uptake into iron-containing droplets. The rate constants of these loss reactions with the $O_2^-$ radical anion are about three orders of magnitude higher than those with

the undissociated $HO_2$ radical (R27, R28, R35-R38 in Table S1). This results in the subsaturation of $HO_2/O_2^-$ in the aqueous phase, i.e. decreasing $\epsilon$ with increasing pH (Fig. 3c). This effect is more pronounced in aerosol water than in cloud water because of the higher Fe concentration in particles. The highly efficient loss of $O_2^-$ at high pH results in the lowest $HO_2/O_2^-$ concentrations in the iron-containing droplets at pH = 6 (Fig. S2d). Since the reactions of Fe(III) with $HO_2/O_2^-$ are the main reduction processes of Fe(III), this low $HO_2/O_2^-$ concentration leads to inefficient Fe(III) to Fe(II) conversion and relatively

higher Fe(III) concentrations. This is reflected in the lower predicted Fe(II)/Fe(total) ratio at low $F_{N,Fe}$ as compared to the FeBulk approach (Fig. 6). The difference in this ratio between $F_{N,Fe}$ = 2% and 100% is about a factor of 2 for cloud water and up to a factor of 3 in aerosol particles. Consequences of this finding for model and field studies are discussed in Section 4.1.

### 3.3 Difference in aqueous phase concentrations and phase transfer rates as a function of $F_{N,Fe}$

#### 3.3.1 Aqueous phase concentrations

The two values of $F_{N,Fe}$ = 2% and 100% likely represent extreme values for the iron distribution among cloud droplets or particles. While, depending on abundance and proximity of iron emissions sources, also fewer particles (and thus droplets) may contain iron, this would translate into an even higher iron aqueous phase concentration. Under such conditions iron would not be completely dissolved and available for aqueous phase reactions. Therefore, in the following, we limit our discussion to values of $F_{N,Fe} \geq 1\%$. In order to illustrate the total ROS budget in the aqueous phase, Fig. 7 shows the bulk aqueous phase

concentration $[ROS]_{aq,bulk}$ [M] calculated based on the LWC-weighted average ROS concentration in the iron-containing and iron-free droplets or particles:

$$[ROS]_{aq,bulk} = [ROS]_{ironfree} \cdot (1 - F_{N,Fe}/100\%) + [ROS]_{iron} \cdot F_{N,Fe}/100\% \qquad (E.5)$$

whereas $[ROS]_{ironfree}$ and $[ROS]_{iron}$ are the ROS concentrations in the iron-free and iron-containing droplets, respectively. (Note that the same calculation was performed for the bulk aqueous phase concentrations in Figs. 2, 4, S1 and S2). Results are shown

for the full range of $1\% \leq F_{N,Fe} \leq 100\%$ and total iron concentrations of 10 ng m$^{-3}$ and 50 ng m$^{-3}$.

The $H_2O_2$ concentrations do not show any dependence on either iron distribution ($F_{N,Fe}$) or iron mass ($m_{Fe}$) (Fig. 7a and d). As in cloud water the main chemical source and loss processes are independent of iron, this finding is obvious. However, in aerosol water, both the main chemical source and loss reactions are dependent on the Fe(II) concentration. Thus, a higher iron mass concentration increases both rates by the same factor, resulting in nearly equal $H_2O_2$ concentrations for both $m_{Fe}$.

The OH concentration in cloud water is underestimated by FeBulk by up to a factor of $\sim 5$ (pH = 6, Fig. 7b); the deviations are only slightly higher for $m_{Fe}$ = 50 ng m$^{-3}$ than for 10 ng m$^{-3}$. The large difference at the highest pH is the consequence of the more efficient OH formation at pH = 6 by the reaction of $O_2^-$ with $O_3$ (Section 3.2). Since in the FeN<100 approach the $HO_2/O_2^-$ concentration is about one order of magnitude higher at this pH value than for FeBulk, the OH concentration is





generally underestimated by assuming $F_{N,Fe}$ = 100%. It can be concluded that in clean conditions, i.e. when the pH value of
cloud water is $\geq\sim$ 5, even for relatively low iron mass concentrations, iron distribution should be taken into account to obtain
the correct OH concentrations in cloud water. In aerosol water, the OH concentration is generally overestimated by FeBulk by
factors of $\leq\sim$ 2 and $\sim$5 for $m_{Fe}$ = 10 ng m$^{-3}$ and $m_{Fe}$ = 50 ng m$^{-3}$ (Fig. 7e), respectively. As discussed in the previous section,
the lower OH concentration in the FeN<100 approach is a consequence of the lower Fe(II) concentration (Fig. 6b) which leads
to less OH formation by the Fenton reaction.

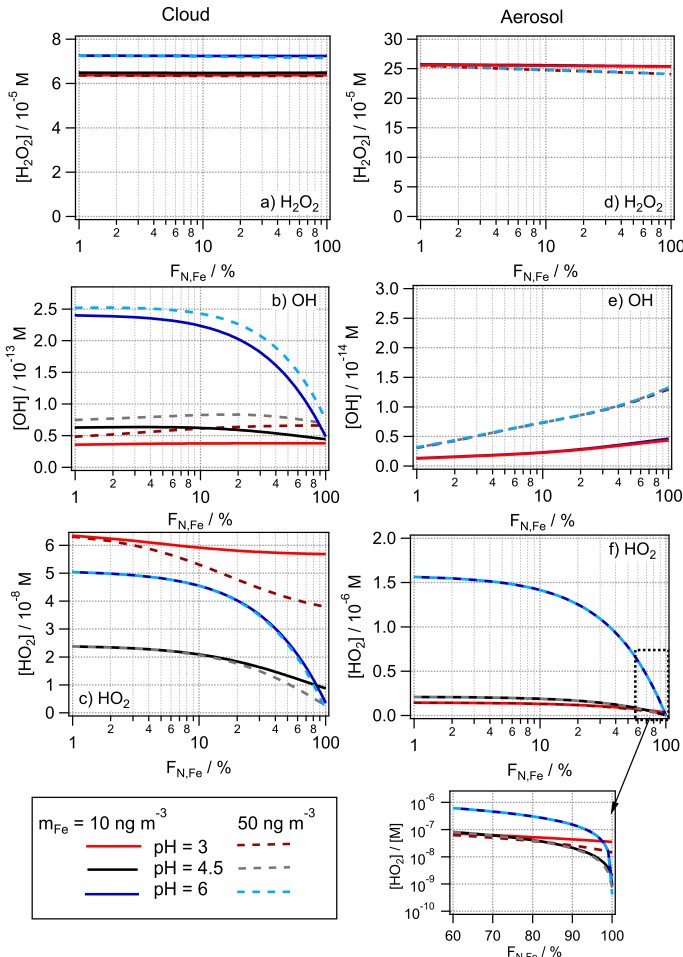

**Figure 7.** Bulk aqueous phase concentrations of H$_2$O$_2$, OH, and HO$_2$ for the cloud (a - c) and aerosol (d - f) cases as a function of $F_{N,Fe}$.
Solid lines are results for $m_{Fe}$ = 10 ng m$^{-3}$, dashed lines for $m_{Fe}$ = 50 ng m$^{-3}$





The $HO_2/O_2^-$ concentrations are underestimated in cloud water by the FeBulk approach, approximately by the same factors ($\sim$2 - 5) as the OH radical, with the highest bias at pH = 6. At pH = 3, the difference between the $HO_2/O_2^-$ concentrations for the two iron mass concentrations is much larger than for the other pH values. As the total iron concentration in the iron-containing particles is five times higher for the higher $m_{Fe}$, this leads to even less Fe(III) reduction (Fe(II)/Fe(total) $\sim$ 0.25 for $F_{N,Fe}$ = 1%, $m_{Fe}$ = 50 ng $^{-3}$) resulting in slightly lower Fe(II)/Fe(total) ratios at all pH values as compared to the lower $m_{Fe}$

(6). With increasing $F_{N,Fe}$ (decreasing iron concentration per particle), the Fe(II)/Fe(total) ratios become similar for both $m_{Fe}$ ($\sim$0.65). The same Fe(II)/Fe(total) ratio at high $F_{N,Fe}$ implies that the Fe(II) concentration scales with $m_{Fe}$ which leads to more efficient $HO_2/O_2^-$ loss and lower $HO_2/O_2^-$ concentrations at high $m_{Fe}$.

    In aerosol water, the aqueous phase phase concentrations for both $m_{Fe}$ seem nearly identical for wide ranges of $F_{N,Fe}$. However, the enlarged figure for $F_{N,Fe} \geq 60\%$ reveals that the aqueous phase concentrations drop by several orders of magnitude

with increasing $F_{N,Fe}$ for pH $\geq$ 4.5. Under more acidic conditions (pH = 3), the difference is much smaller (factor of $\leq$ 10 across the $F_{N,Fe}$ range). Thus, the aqueous phase budget of $HO_2/O_2^-$ may be underestimated by several orders of magnitude with increasing pH if the iron distribution across the particle distribution is not properly accounted for.

### 3.3.2   Relative difference in phase transfer rates $\Delta R_{PT}[ROS]$

    As discussed in Sections 3.1 and 3.2, the ROS phase transfer rates are a consequence of an imbalance between the chemical

source and loss rates in the aqueous phase. To quantify the differences in the net phase transfer rates, we define

$$\Delta R_{PT}[ROS] = \left( \frac{R_{FeN<100}}{R_{FeBulk}} - 1 \right) \cdot 100\% \tag{E.6}$$

whereas $R_{FeN<100}$ and $R_{FeBulk}$ are the net phase transfer rates in the two model approaches. Figure S3 shows $\Delta R_{PT}[ROS]$ as a function of $F_{N,Fe}$ for the ROS and cloud and aerosol cases; by definition, $\Delta R_{PT}[ROS]$ is zero for $F_{N,Fe}$ = 100%. A positive $\Delta R_{PT}[ROS]$ mans that the phase transfer rate is underestimated by the FeBulk approach.

Similar to the differences in ROS aqueous phase concentrations in Fig. 7, large differences in $H_2O_2$ phase transfer rates are predicted for cloud droplets at pH = 6 because of the inefficient $H_2O_2$ consumption in iron-free droplets. In aerosol water, the phase transfer rates of $H_2O_2$ differ by up to a factor of $\sim$2 ($\Delta R_{PT}[H_2O_2] \geq$ -100%). However, since the $H_2O_2$ uptake is sufficiently efficient to maintain thermodynamic equilibrium, such differences in $H_2O_2$ uptake rates may be negligible.

    For the OH radical, the highest discrepancy between the uptake rates is predicted at pH = 4.5 ($\leq$ 30 - 50%, depending on $m_{Fe}$,

Figure S3b). For FeBulk, the OH concentration in clouds is nearly independent of pH ($3.9 \cdot 10^{-14}$ M $\leq$ [OH] $\leq 4.5 \cdot 10^{-14}$ M, Figs. 2 and S2). However, for FeN<100, the OH concentration is increasing with pH in the iron-free droplets whereas it is slightly decreasing in the iron-containing droplets. These opposite trends are due to the non-linear rate of the $HO_2/O_2^-$ recombination, that is highest at pH = 4.5 and dominating OH formation in the iron-free droplets whereas in the iron-containing droplets the pH-independent Fenton reaction is the most important OH source (Fig. 5b). In aerosol, $\Delta R_{PT}[OH]$ are even smaller, on the

order of < 5%, even though OH is not in thermodynamic equilibrium ($\epsilon$ < 1, Fig. 3b).

    The relative differences in $HO_2$ for $m_{Fe}$ = 10 ng m$^{-3}$ are < 20% for all conditions in cloud water (Fig. S3c). The highest value is predicted at pH = 3 and $m_{Fe}$ = 50 ng m$^{-3}$ as a consequence of the increasing difference of the $HO_2$ formation rate by the





Fe(II) reaction with molecular oxygen and its loss rate via its reaction with Fe(III) (Fig. 5c) and the low Fe(II)/Fe(total) ratio.

For the aerosol case, the phase transfer rates of $HO_2$ differ by up to a factor of $\sim 2$ ($\Delta R_{PT}[HO_2] \geq$ -100%), depending on $F_{N,Fe}$,

with the smallest deviation at pH = 6 ($\Delta R_{PT}[HO_2] \geq$ -60%). At high pH, the phase transfer is relatively more important as $HO_2$ source ($\leq 20\%$) as compared to the more acidic conditions (Fig. 5f). The evaporation of $HO_2$ at pH = 3 from the iron-free particles reduces the net uptake rate of $HO_2$ into the particle phase. Therefore, the difference in the net uptake rates are largest at low pH value.

     The uptake of OH and $HO_2$ into aerosol particles is often parameterized by the dimensionless reactive uptake coefficient $\gamma$

(Thornton and Abbatt, 2005; Mao et al., 2013, 2017) that is derived based on the first-order radical loss ($k^{loss}$) onto aerosol particles:

$$k^{loss} = \gamma \frac{\omega S}{4} \tag{E.7}$$

with $\omega$ being the molecular speed,

$$\omega = \sqrt{\frac{8RT}{\pi M}} \tag{E.8}$$

whereas M is the molar mass [g mol$^{-1}$], R is the gas constant, and T is the absolute temperature [K], and S the aerosol surface, e.g., Thornton and Abbatt (2005); Pöschl et al. (2007). Accordingly the phase transfer rate of the radicals into the particle phase can be described as

$$\frac{d[Radical]_g}{dt} = k^{loss}[Radical]_g \tag{E.9}$$

     Equation E.9 corresponds to the first right-hand term of Equation E.2 and thus, $\gamma$ values for OH and $HO_2$ were derived based

on the phase transfer rates in Figure S3 (Fig. 8). We do not show values for $H_2O_2$ since its partitioning can be described based on Henry's law.

     The $\gamma$ values for $HO_2$ into cloud water are in the range of $10^{-3}$- $10^{-2}$ with higher values at the highest pH. The weak $F_{N,Fe}$ dependence of the total phase transfer rates into cloud water (Fig. S3c) is also reflected in its $\gamma$ values with the highest value at pH = 6, when also the phase transfer rate is highest. The $\gamma(OH)$ values show even less difference between different pH values

and $F_{N,Fe}$ (note the linear scale in Fig. 8a). Thus, the reactive uptake of OH into cloud droplets could be parameterized by a constant value of $\gamma \sim 6 \cdot 10^{-3}$ for all conditions explored here.

     In aerosol water, $\gamma(HO_2$ shows a strong dependence on $F_{N,Fe}$, leading to an overestimate of the reactive uptake by FeBulk by up to two orders of magnitude at pH = 3 (Fig. 8c). Note that at $F_{N,Fe}$ = 1% no $\gamma$ value is shown as under such conditions, $HO_2$ is predicted to evaporate. This evaporation is due to the inefficient $HO_2$ consumption in iron-free particles but efficient uptake

into the iron-containing particles,independent of the aqueous phase concentration. This leads to a great imbalance of the phase transfer rates from and into the two droplet classes than shown in Fig. 2b. Similar to the findings for cloud water, the uptake of the OH radical could be parameterized by a single $\gamma$ value for the conditions applied here ($\gamma \sim 0.0045$).


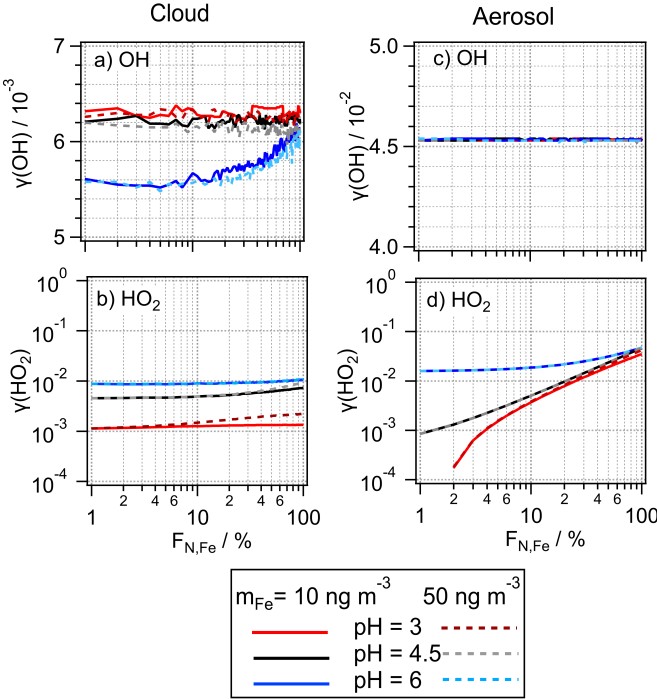

**Figure 8.** Reactive uptake coefficients $\gamma(OH)$ and $\gamma(HO_2$ derived from the multiphase model studies at three pH values for a, b) cloud, c, d) aerosol conditions. The solid and dashed lines show results for $m_{Fe}$ = 10 and 50 ng m$^{-3}$, respectively.

### 3.3.3 Enhanced $H_2O_2$ partitioning into aerosol water, $K_{H,eff,H2O2}$ = 2.7·10$^8$ M atm$^{-1}$

Strictly, Henry's law constants are not applicable to highly concentrated aqueous solutions such as aerosol water. Simultaneous

measurements of particle-bound and gas phase $H_2O_2$ concentrations suggest that it partitions much more efficiently to the

particle phase than predicted on its physical Henry's law constant ($K_{H,H2O2}$ = 1.02·10$^5$ M atm$^{-1}$). Despite large uncertainties in

such measurements and of related parameters such as the aerosol water content, various studies revealed that the partitioning

of $H_2O_2$ between the gas and the aerosol aqueous phases may be more appropriately described with an effective Henry's law

constant of $K_{H,eff,H2O2} \leq 2.7\cdot10^8$ M atm$^{-1}$ (Hasson and Paulson, 2003; Xuan et al., 2020). The observation by Hasson and

Paulson (2003) of higher $H_2O_2$ concentrations in coarse mode particles than in fine mode particles might point to a size-

dependent chemical particle composition with higher iron content in coarse particles that often contain dust.

To explore the effects of such enhanced partitioning, we performed a sensitivity study, using $K_{H,eff,H2O2} \leq 2.7\cdot10^8$ M atm$^{-1}$.

In Figure S4, the aqueous phase concentrations and net phase transfer rates of the three ROS are compared to those using

the physical Henry's law constant ($K_H(H_2O_2)$ = 1.02·10$^5$ M atm$^{-1}$). No other parameter was changed as to our knowledge

corresponding $K_{H,eff}$ for OH and/or $HO_2$ are not available. We perform this comparison only for the aerosol case, as numerous



measurements have shown that the partitioning of $H_2O_2$ into cloud water can be satisfactorily described by its $K_H$ (e.g., Ervens (2015)).

As expected, a higher ($K_{H,eff,H2O2}$) results in higher $H_2O_2$ aqueous phase concentrations by three orders of magnitude. Also the phase transfer of $H_2O_2$ into the particle phase is more efficient even though the difference in the rates is smaller than the

ratio of the $K_{H(eff)}$ values. However, unlike for the low $K_{H,H2O2}$, the higher partitioning leads to smaller $H_2O_2$ concentrations for FeBulk than for FeN<100 by up to one order of magnitude and pH = 6. The OH concentration in the aqueous phase is predicted to decrease with increasing $F_{N,Fe}$, unlike in our base case simulation where we showed the opposite trend (Fig. S4b). The higher $H_2O_2$ concentration generally increases the importance of the Fenton reaction and photolysis leading to more OH production in the aqueous phase. While the $HO_2/O_2^-$ concentrations for the two $K_{H,H2O2}$ values do not differ at low $F_{N,Fe}$,

they are significantly smaller by several orders of magnitude for the higher $K_{H,H2O2}$ (Figure S4c). For the full range of $F_{N,Fe}$, the Fe(II)/Fe(total ratio does not exceed ∼20% (Fig. S4g). Thus, $HO_2/O_2^-$ is always efficiently consumed by Fe(III). This predicted independence of Fe(II)/Fe(total) ratio as a function of $F_{N,Fe}$ for the high $K_{H,eff,H2O2}$ in contrast to the predicted strong dependence by using the physical $K_{H,H2O2}$ may be used to guide future experiments to determine the 'best' value of $K_{H,eff,H2O2}$ for aerosol water.

The predicted phase transfer rates for OH and $HO_2$ are nearly the same for both $K_{H(eff),H2O2}$ at $F_{N,Fe} > \sim 10\%$ which implies that the derived $\gamma$ values for these radicals might apply for both $K_{H(eff),H2O2}$ values. However, at $F_{N,Fe} < 10\%$, we predict a net evaporation of $HO_2$ from the particle phase ($R_{PT}HO_2 < 0$, Fig. S4f). This implies that the particle phase is supersaturated in $HO_2/O_2^-$. Such considerations need to be taken into account in studies of the ROS formation potential of particles in the context of oxidative properties and health effects of particulate matter, e.g. (Saffari et al., 2014; Tong et al., 2016, 2020) (Section 4.3).

## 4    Implications of $F_{N,Fe} < 100\%$ for model and field studies

### 4.1    Oxidation state of iron: Fe(II)/Fe(III)

The iron oxidation state determines its solubility and bioavailability since Fe(II) salts are usually more soluble than Fe(III) compounds. Fe(II) usually dominates during daytime because Fe(III) complexes are readily photolysed (Deguillaume et al., 2005). Reported measurements are based on analyses that are performed on bulk samples where redox reactions of iron can oc-

cur after collection and combination of all cloud droplets or aerosol particles, respectively. The trends and overall Fe(II)/Fe(III) ratios have been reproduced in several model studies that all assumed that iron is present in the complete aqueous phase (Ervens et al., 2003; Deguillaume et al., 2004; Luo et al., 2005; Tilgner et al., 2013; Ingall et al., 2018). Thus, the agreement between measured and predicted data is not surprising as both types of studies imply that the total soluble iron is evenly distributed throughout the total aqueous volume.

However, our study reveals that these Fe(II)/Fe(III) ratios are not the true ratios present in the atmospheric aqueous phases if iron is only present in a small number fraction of particles or droplets. In cloud water, the soluble iron fraction ($\sim$Fe(II)/Fe(total)) might be overestimated by up to factor of 2 (Fig. 6a) whereas this bias might be even higher in aerosol particles ($\leq\sim$2.5 (Fig. 6b). The overview articles by Deguillaume et al. (2005) and Mao et al. (2017) show large ranges of the Fe(II)/Fe(total)





fraction (10 - 100%). Interestingly the study by Takahama et al. (2008) that investigated single particles report values at the
lower end of this range (∼20%). We cannot state with certainly that this finding is due to different aerosol composition or
origin or indeed points to a more accurate representation of the Fe(II)/Fe(II) ratio in individual particles rather than in bulk
samples.

If the Fe(II))/Fe(total) may be considered a measure of the bioavailability of iron, it might be actually lower than derived
from ambient samples or model studies, if they apply a bulk approach. Our results suggest that parameterizations of the
Fe(II)/Fe(total) ratios as, for example suggested by (Mao et al., 2017), might have to be further refined to account for the
effects due to the iron distribution across particle populations. In aqueous media that consist of a bulk aqueous phase (e.g.,
oceans, lung fluid, to some extent also rain water), the Fe(II) fraction will be appropriately represented by FeBulk ($F_{N,Fe}$ =
100%). The time scales for the Fe(II)/Fe(III) ratio to adjust from conditions of $F_{N,Fe} < 100\%$ to $F_{N,Fe} = 100\%$ will depend on
the rates of iron redox processes, and on the solubility and dissolution kinetics of ferrous and ferric salts which in turn maybe
a function of other solutes and liquid water content.

## 4.2 Application of reactive uptake parameters $\gamma_{OH}$ and $\gamma_{HO2}$

### 4.2.1 Reactive uptake coefficient of the hydroxy radical, $\gamma(OH)$

The values for $\gamma(OH)$ derived based on our model studies for both iron mass concentration and over the full range of $F_{N,Fe}$
($5.8 \cdot 10^{-3} \leq \gamma(OH)_{cloud} \leq 6.4 \cdot 10^{-3}$; $\gamma(OH)_{aerosol} \sim 4.5 \cdot 10^{-3}$ are in good agreement with the average value suggested for pure
water surfaces ($\gamma = 0.035$) (Hanson et al., 1992). It should be noted that this value was derived assuming a Henry's law constant
of $K_H(OH) \geq 40$ atm M$^{-1}$ and $[OH] \geq 10^8$ cm$^{-3}$. If it were adjusted to atmospherically more relevant conditions ($[OH] \sim 10^6$
cm$^{-3}$) and the $K_H(OH) = 25$ M atm$^{-1}$ as used in our study (Table S4), the resulting $\gamma(OH)$ might be smaller by more than order
of magnitude than the reported one. Molecular dynamics simulations for the mass accommodation of OH on water surfaces
showed $\alpha(OH) = 0.1$ which can be considered the upper limit of reactive OH uptake onto water surfaces (Roeselová et al.,
2003). The difference between such a value for pure water and the values from our model study demonstrate the high reactivity
of the OH radical in the aqueous phase. As its solubility is limited, this high reactivity results in subsaturated conditions ($\epsilon(OH)$
< 1; Fig. 3b).

Several previous studies have determined $\gamma(OH)$ on surfaces other than pure water that are more comparable to atmospheric
aerosol particles. Hanson et al. (1992) showed that $\gamma(OH)$ increases in acidic solutions from 0.07 (28% H$_2$SO$_4$) to unity in
96% H$_2$SO$_4$. On organic surfaces, $\gamma(OH)$ values are generally higher than on inorganic surfaces which demonstrates the high
reactivity of OH with organics in the condensed phase and/or on organic surfaces. In several lab studies, squalane was used as a
proxy for long-chain alkane; the reactive uptake coefficients have been determined in a range of $\sim 0.2 < \gamma(OH) < 0.5$ (e.g., Che
et al. (2009); Smith et al. (2009); Waring et al. (2011); Houle et al. (2015); Bianchini et al. (2018); Li and Knopf (2021)). A
similar value range was also found for a wide variety of other organics ((Bertram et al., 2001)). In other experimental studies on
organic particles, $\gamma(OH) > 1$ were found ($\gamma(OH) = 1.64$ on wet succinic acid (Chan et al., 2014), $1.3 \pm 0.4$ on bis(2-ethylhexyl)
sebacate (George et al., 2007)). $\gamma(OH)$ on organic aerosols may decrease with increasing OH gas phase concentration because





the viscosity of the organic condensed phase prevents efficient uptake and diffusion of OH towards the particle center (Slade and Knopf, 2013; Arangio et al., 2015). Such effects were included in recently developed frameworks to parameterize $\gamma(OH)$ as a function of particle viscosity, size and gas phase OH concentration (Renbaum and Smith, 2011; Houle et al., 2015).

We did not explore any of these parameters in detail in our current model study. However, the comparison of the tight range of our $\gamma(OH)$ values in Fig. 8, independently of $F_{N,Fe}$ and cloud or aerosol aqueous conditions, to the wide ranges of literature values, leads us to the following conclusions: (i) The similarity of our $\gamma$ value based on an explicit chemical mechanism to that measured on pure water suggests that $\sim 10^{-3} < \gamma(OH) \sim 10^{-2}$ may be a good approximation for cloud conditions. (ii) The fact that we cannot reproduce the high $\gamma(OH)$ values as observed on organic aerosols suggests that organic reactions control

the reactive uptake of OH which are not included in our explicit chemistry scheme. (iii) Since organics are likely present in all aerosol particles, the effect of the iron distribution ($F_{N,Fe}$) for $\gamma(OH)$ may be overestimated by our chemical mechanism on ambient aerosol particles.

### 4.2.2   Reactive uptake coefficient of the hydroxy peroxyl radical, $\gamma(HO_2)$

It has been shown in global model studies, that the the reactive loss of $HO_2$ on aerosol surfaces can significantly impact the

atmospheric oxidant budget, e.g. (Haggerstone et al., 2005; Macintyre and Evans, 2011; Stadtler et al., 2018; Li et al., 2019). A value of $\gamma(HO_2) = 0.2$ based on the study by Jacob (2000) has been commonly used in global models, which has also been supported by molecular dynamics simulations (Morita et al., 2004). However, recent model studies suggested that this value may be an overestimate and that values on the order of $\gamma(HO_2) = 0.05$ or 0.08 may be more appropriate (Christian et al., 2017; Tan et al., 2020).

On dry aerosol surfaces, $\gamma(HO_2)$ is usually low: For example, Bedjanian et al. (2005) measured $\gamma(HO_2) = 0.075 \pm 0.015$ on dry soot surfaces which is comparable to the value of $\gamma(HO_2) \leq 0.031$ on dust (Matthews et al., 2014) and $\gamma < 0.01$ on dry NaCl (Remorov et al., 2002). $\gamma(HO_2)$ increases by more than one order magnitude when RH increases from $\sim 20\%$ to $> 90\%$ (Cooper and Abbatt, 1996; Taketani et al., 2008; Lakey et al., 2015, 2016; Moon et al., 2018). Lakey et al. (2015) showed much higher $\gamma(HO_2)$ values on two humic acids ($0.0007 \leq \gamma(HO_2) \leq 0.06$ and $0.043 \leq \gamma(HO_2) \leq 0.09$, respectively) than on pure

long-chain carboxylic acids ($\gamma(HO_2) < 0.004$). They ascribed the higher $HO_2$ uptake to the small concentrations of copper and iron ions in the humic acids, in agreement with previous studies that showed the large impact of TMI ions on $HO_2$ reactive uptake (e.g., (Mao et al., 2013, 2017)). It was suggested by (Lakey et al., 2015) that the $HO_2$ uptake cannot be solely controlled by the TMI ion concentration as they did not see any correlation with $\gamma(HO_2)$. This lack of trend is similar to findings by Mozurkewich et al. (1987) who also did not find any clear relationship in $\gamma(HO_2)$ on Cu-doped $NH_4HSO_4$ and $LiNO_3$ particles

with Cu concentration.

Based on our model results, the lack of such correlation can be qualitatively understood by simultaneous increase of the main chemical $HO_2$ source and sink rates (Reactions S1 and L4- L6 in Fig. 5f). The absence of a dependence of $HO_2$ uptake on wet particle diameter in the study by Mozurkewich et al. (1987) is also in agreement with our sensitivity studies in which we varied the particle diameter by a factor of $\pm 2$ (Fig. S6). This implies that the interfacial mass transfer is not a limiting step in the $HO_2$

uptake. The $HO_2$ uptake on Cu-doped ammonium sulfate particles was observed to be higher under neutral ($\gamma(HO_2) = 0.2$) as

opposed to acidic conditions ($\gamma$(HO$_2$) = 0.05) (Thornton and Abbatt, 2005). Similar results were also observed by Mao et al. (2013) who generally found somewhat higher values than in our study ($\gamma$(HO$_2$) > 0.4) and highest values for lowest Cu/Fe ratios (i.e., highest iron concentration at a constant copper concentration) and highest pH. However, overall their differences in $\gamma$(HO$_2$) due to Fe/Cu ratio and/or pH are smaller than those that we predict for different $F_{N,Fe}$ ($\sim$10$^{-4}$ $\leq$ $\gamma$(HO$_2$) $\leq$ $\sim$10$^{-2}$, at
pH = 3). This suggests that $\gamma$(HO$_2$) parameterizations that take into account the dependence on Cu concentration, total aerosol mass and particle size e.g., Guo et al. (2019); Song et al. (2020), should be further refined to also include the iron distribution.

    We note that our results are not quantitatively comparable to those on Cu-doped particles as our chemical scheme includes only reaction of iron as the only TMI. The corresponding reactions with Cu$^{+/2+}$ have generally higher rate constants than those of iron reactions (Ervens et al., 2003; Deguillaume et al., 2004); however, as copper concentrations in ambient aerosols are
usually lower than iron concentrations, the effect of both TMIs might be similar. We did not perform the equivalent simulations for copper in the present study as to our knowledge, there is no data available that give single-particle information on the distribution of iron and copper within droplet or particle populations. However, if iron and copper originated from different emission sources, the number concentration of TMI-containing particles might be relatively high. However, in such situations the synergistic effects of Fe and Cu affecting HO$_x$ cycling and uptake as suggested by (Mao et al., 2013) might be reduced.

Measurements of $\gamma$(HO$_2$) on ambient aerosol populations showed ranges of 0.13 $\leq$ $\gamma$(HO$_2$) $\leq$ 0.34 (Mt. Tai) and 0.09 $\leq$ $\gamma$(HO$_2$)$\leq$ 0.4 (Mt. Mang) Taketani et al. (2012). No systematic trend with total iron concentration that exceeded 1 $\mu$g m$^{-3}$ at Mt Mang and 10 $\mu$g m$^{-3}$ at Mt Tai was observed. The acidity of the aerosols was not reported in that study; however, the derived $\gamma$(HO$_2$) values did not show any significant trend with the molar NH$_4^+$/(SO$_4^{2-}$ + NO$_3^-$) ratios that ranged from $\sim$1 - $\sim$2 and $\sim$2 - $\sim$3, pointing to moderately acidic to fully neutralized aerosol. The fact that the ambient data do not show any
trend with acidity might either point to a high number fraction of TMI-containing particles and/or to a relatively high pH, i.e. to conditions under which our model results do not show a large effect of $F_{N,Fe}$ (Fig. 8d). Two studies of ambient aerosol at urban locations revealed similar average values as compared to the mountain sites with $\gamma$(HO$_2$) = 0.23 $\pm$ 0.22 in Yokohama, Japan (Zhou et al., 2021) and 0.24 (0.08 $\leq$ $\gamma$(HO$_2$) $\leq$ 0.36) in Kyoto, Japan (Zhou et al., 2020). No significant trend in the $\gamma$ values was observed as a function of aerosol or air mass characteristics (coastal or mainland).

Given the large ranges of $\gamma$(HO$_2$) values on lab-generated and ambient aerosol, it is difficult to reconcile differences between lab data and observations with our model results. However, our results might provide an explanation for the discrepancies of lab-derived or theoretical $\gamma$(HO$_2$) values in the range of 0.2 - 1 to those determined on ambient aerosol (< 0.4). We cannot say for sure that these differences indeed stem from different assumptions of iron distribution or other factors affecting $\gamma$(HO$_2$). However, our strong predicted decrease of $\gamma$(HO$_2$) values at low pH and $F_{N,Fe}$ suggests that applying lab-derived $\gamma$(HO$_2$) on
aerosol particles with identical composition might lead to an overestimate of HO$_2$ loss on atmospheric aerosol surfaces, even if all other aerosol properties (size, total iron mass, pH etc) are identical.

### 4.3 ROS budgets and oxidative potential (OP)

The formation and cycling of ROS in the aqueous phase is tightly linked to adverse health effects of particulate matter that are caused by oxidative stress in the respiratory tract and lungs (Stohs and Bagchi, 1995; Tong et al., 2017; Molina et al., 2020;





Shahpoury et al., 2021). In addition, the oxidant content of ambient aerosol particles controls their ageing time scales and
interactions with the biosphere (Pöschl and Shiraiwa, 2015).

    Tong et al. (2020) found a positive trend of reactive species concentrations with total particle mass and increased OH concentrations as a function of transition metal content and the opposite trend for organic radicals at remote forest and polluted urban locations . Contrarily, Verma et al. (2012) did not find any correlation of the ROS formation potential with metal concentrations.

Fang et al. (2016, 2017) suggested that sulfate may trigger metal solubility in highly acidic supermicron particles, which in turn leads to enhanced ROS formation. On a molar basis quinones and TMIs may be equally efficient in producing ROS (Charrier and Anastasio, 2012; Lyu et al., 2018); however, as TMI concentrations are higher than those of quinones in aerosol water, TMIs are considered major drivers for ROS levels. In the overview study by Saffari et al. (2014) ROS activity was compared at different locations; at nearly all locations highest activity was observed in $PM_{2.5}$ (as opposed to $PM_{10}$) samples; this trend

was explained with the higher fractions and solubilities of iron in particles of these sizes.

    Based on such studies, the oxidative potential has been related to the presence and amount of TMI in aerosol particles. Our model studies suggest that the total iron amount within an aerosol population may not be a sufficient parameter to constrain the oxidant concentrations within particulate matter. $H_2O_2$ is often considered one of the main contributors to the total oxidant budget in particles (Tong et al., 2016). Given the large uncertainty in the partitioning of $H_2O_2$ into aerosol water (Section

3.3.3), the extent is not clear to which the iron distribution affects its total concentration in the particles phase. The $H_2O_2$ budget may be overestimated by up to an order of magnitude if the enhanced Henry's law constant ($K_{Heff,H2O2} = 2.7 \cdot 10^8$ M atm$^{-1}$) is more appropriate (Fig. S4). We find that the OH concentration in particles can be over- or underestimated at low $F_{N,Fe}$, depending on $K_{Heff,H2O2}$ (Fig. S4b), whereas the $HO_2$ may be overestimated if $K_{Heff,H2O2}$ is higher than the physical Henry's law constant (Fig. S4c). The partitioning of $H_2O_2$ into aerosol water (i.e. $K_{Heff,H2O2}$) is likely dependent on the aerosol

composition. Therefore, this dependence together with the [ROS] dependence on $F_{N,Fe}$ might explain the different conclusions regarding the trends of ROS budgets in particles of different air masses. Similar to the Fe(II)/Fe(III) ratio, our model results suggest that bulk samples of aerosol particles may not accurately represent the ROS budget in ambient populations. Upon sampling, the particles might get dissolved and undergo the ROS formation and conversion processes. Several studies have suggested efficient ROS formation from secondary organic material (e.g., Tong et al. (2018); Zhou et al. (2019); Wei et al.

(2021)). The underlying formation processes are not fully clear; given that SOA is likely present in all articles (and droplets), the effect of iron distribution on ROS formation might become less important if ROS yields from purely organic particles are sufficiently high. However, it has been shown that ROS formation by SOA can be enhanced in the presence of metal ions (Nguyen et al., 2013; Tuet et al., 2017).

    While the ROS budgets of atmospheric particulate matter might be different depending on these parameters, these uncertain-

ties might be diminished upon inhalation: Firstly, all iron will be dissolved such that the solution can be described by FeBulk. Secondly, the reaction medium is limited to a bulk liquid phase (lung fluid) and thus $H_2O_2$ gas/aqueous partitioning does not play a role.

    Generally all conclusions for oxidant budgets in aerosol water also apply to those in cloud water. However, the largest concentration differences for extreme values of $F_{N,Fe}$ are less than an order of magnitude (Section 3.1). Several model studies



have explored the sources and production rates of OH in cloud water, e.g., Ervens et al. (2003); Deguillaume et al. (2004); Tilgner et al. (2013); Bianco et al. (2015). However, our analysis shows that the chemical source and loss rates might be underestimated in bulk cloud water. While the overall oxidant content of the cloud water might not be significantly affected ('bulk aqueous phase concentrations' in Fig. 2), the large differences of OH concentrations of about an order of magnitude in iron-containing and iron-free droplets may lead to very different oxidation rates of organics. Higher OH concentrations in

the aqueous phase may also lead to more efficient formation of organics acids (aqSOA; Ervens et al. (2011, 2014). Generally highest OH concentrations in cloud water are predicted under clean (marine, remote) conditions (Herrmann et al., 2000; Tilgner et al., 2013). Our results suggest that the difference in OH concentrations might be even greater between low and high pH as we find the largest impact by $F_{N,Fe}$ for high pH. Even though in clean air masses, total iron concentrations might be lower, such trends might be robust as we do not find a strong dependence of the predicted OH concentrations on $m_{Fe}$. Given the close

connections of OH and $HO_2/O_2^-$, any conclusions for the OH concentrations in cloud water can be similarly drawn also for $HO_2/O_2^-$. In contrast, the less reactive and more soluble $H_2O_2$ is not affected by the iron distribution.

## 5 Summary and conclusions

The role of transition metal ion (TMI) reactions for impacting oxidant levels (OH, $HO_2$, $H_2O_2$) in the atmospheric aqueous phase has been recognized for a long time. However, in atmospheric multiphase chemistry models, it usually assumed that all

aerosol particles and droplets contain TMI and thus TMI catalyzed reaction occur in all droplets. Single-particle analyses have shown that only a small number fraction of particles contain iron which implies the same for cloud droplets that are formed on such particles.

Using a box model with a well-established chemical multiphase, we explored the importance of iron distribution across (i) aqueous aerosol particle or (ii) cloud droplet populations. We performed box model studies in which a constant iron concen-

tration (10 or 50 ng m$^{-3}$) is distributed to a number fraction $F_{N,Fe}$ of aerosol particles or droplets from 1% to 100% (FeN<100 and FeBulk approaches respectively).

We find that $H_2O_2$ concentrations in cloud water are not affected by $F_{N,Fe}$ as $H_2O_2$ is in thermodynamic equilibrium between the gas phase and all droplets. The same conclusions are drawn for $H_2O_2$ partitioning into aerosol water if the same Henry's law constant is applied ($K_{H,H2O2} = 1.02 \cdot 10^5$ M atm$^{-1}$). However, since $H_2O_2$ has been found to partitioning more efficiently

into aerosol than into pure water ($K_{Heff,H2O2} = 2.7 \cdot 10^8$ M atm$^{-1}$), We find that $H_2O_2$ concentrations in cloud water might be underestimated by the FeBulk approach whereas it maybe overestimated in aerosol water. These differences are largest at pH = 6 whereas the differences are negligible at pH = 3 or 4.5. The opposite trends are found for the OH radical whose aqueous phase concentration in cloud water might be overestimated by up to a factor of 5 using the FeBulk approach at pH = 6 whereas there might be an over- or underestimate by nearly an order of magnitude in aerosol water, depending on the choice of $K_{Heff,H2O2}$.

The concentrations of OH and $HO_2$ radicals are closely linked. Generally we find that the $HO_2\emptyset_2^-$ concentrations in the aqueous phase are underestimated by the FeBulk approach, again with the largest discrepancies at pH = 6 for $F_{N,Fe} = 1\%$ or 100%. This trend can be explained by the increase of the effective Henry's law constant of $HO_2$ at high pH ($K_{Heff,HO2} = $





$1.5 \cdot 10^5$ M atm$^{-1}$) and the higher rate constants of $O_2^-$ radical anion as compared to the corresponding reactions of the $HO_2$

radical. The higher partitioning together with the quicker consumption of $HO_2\emptyset_2^-$ at high pH leads to its subsaturation in the

aqueous phase at high pH. This effect is strongest at low $F_{N,Fe}$ because the few iron-containing droplets (particles) are highly

concentrated in iron. The reaction of Fe(III) ions or hydroxy complexes with $HO_2\emptyset_2^-$ is the main sink of $HO_2\emptyset_2^-$ in both

the cloud and aerosol aqueous phases. At the same time it is also the main reduction pathway of Fe(III) to Fe(II). Since the

$HO_2\emptyset_2^-$ concentration in iron-containing droplets is up to one orders of magnitude lower in the FeBulk approach, respectively,

the Fe(III) reduction is less efficient. Our results are largely independent form the total iron mass concentration as both the

chemical $HO_2\emptyset_2^-$ sources and sinks are dependent on iron. Based on these results, we conclude that the Fe(II)/Fe(III) ratio

in cloud and aerosol water might be lower than implied by bulk samples. This finding has implications on the interpretation

of modeled and measured Fe(II)/Fe(III) ratios that are often considered a measure of iron solubility and bioavailability. Our

study demonstrates that multiphase chemistry models might not be able to predict properly OH and $HO_2$ concentrations in

the aqueous phase of cloud or aqueous particle which might translate into biases in the predicted oxidation rates (e.g. aqSOA

formation). While not explored in detail in the current study, further implications of iron distribution in cloud droplets might

include differences in sulfate formation rates by metal-catalyzed processes.

The reactive uptake of OH and $HO_2$ is often parameterized by $\gamma$. We derived $\gamma$ values into aerosol water from our model

studies. While $\gamma$(OH) is not largely affected by the iron distribution, $\gamma$($HO_2$) can be significantly reduced at low $F_{N,Fe}$ as

compared to bulk samples, by up to two orders of magnitude at pH = 3 ($\gamma$($HO_2$ ~0.05 for $F_{N,Fe}$ = 100% and $\gamma$($HO_2$ ~0.03

for $F_{N,Fe}$ = 100% and 0.0002 for $F_{N,Fe}$ = 2%). This analysis demonstrates that $\gamma$ values that are measured in lab studies using

internally mixed aerosol particles might bee too high as compared to those ambient aerosol populations. In addition to another

factors that affect $\gamma$($HO_2$) on various aerosol surfaces (chemical composition, particle size, water content etc), our results

suggest that also the iron distribution across particle populations can add to the large variability of $\gamma$($HO_2$) and to the mismatch

of observed and modeled $HO_2$ loss if lab-derived data are used. Similar considerations can be also applied to the interpretation

of ROS formation and oxidative potential of aerosol particles. As ambient aerosol particles likely comprise external mixtures

in terms of iron, neither results based on bulk sampling nor on modeling will correctly represent the oxidant budgets in ambient

particulate matter. Our study is to our knowledge the first explicit chemical multiphase model study that systematically explores

the role of iron distribution across individual aqueous aerosol particles or cloud droplets for the concentrations and uptake rates

of reactive oxygen species. There is only few data available on the iron distribution in aerosol populations form single-particle

analyses, that may be used on constrain the number fraction of particles and droplets that contain iron ($F_{N,Fe}$) for clouds and

aerosols. While we restricted our model studies to iron, similar conclusions may be also drawn for other transition metal ions,

such as copper or manganese. We identified various potential implications of this parameter for e.g. (i) oxidant budgets and

particle mass formation (aqSOA, sulfate) in clouds and aerosols, (ii) reactive radical (OH, $HO_2$) uptake onto aerosols, and (iii)

oxidative potentials of aerosol particles that should be further explored by experimental and model studies.

*Data availability.* Details on model code and model data can be obtained upon request from the authors



*Author contributions.* BE designed and led the study. AK performed the model studies. AK and BE wrote manuscript; MZ contributed to the discussion of the results.

*Competing interests.* The authors declare that they have no conflict of interest

*Acknowledgements.* This work has been supported by the French National Research Agency (ANR) (grant no. ANR-17-MPGA- 0013). We
thank Daniel Murphy (NOAA/ESRL) for useful discussion and sharing data of single particle measurements.



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
