# Peer review of "The number fraction of iron-containing particles affects OH, HO2 and H2O2 budgets in the atmospheric aqueous phase"

_Atmospheric Chemistry and Physics, 2021_

## Author Response (AR1)

**Author response to Comments by both referees:**

We thank both referees for their detailed and constructive comments. We address them in detail below. Referee comments are in black, our responses in blue and manuscript text in *green italics*.

**Comments by Referee 1:**

This is an interesting paper, as the first one to address how iron distribution across aerosols and droplets may affect multiphase chemistry and reactive oxygen species (ROS). The author has done a good job on examining the budget of ROS in different scenarios. The results are interesting and useful to the community. I have several major comments:

**Author Response:** We thank the referee for their positive comments on our manuscript and the constructive suggestions that helped to improve its clarity and structure. We address all individual comments point-by-point below. Figure numbers refer to the new manuscript; line numbers to the manuscript version without track change.

**Referee Comment 1.1:** Aqueous diffusion. Given the short lifetime of ROS in aqueous phase, one would expect a large difference between surface concentration and bulk concentration of ROS for individual aerosols or droplets. This can be addressed by introducing a diffuso-reactive parameter (see Mao et al. 2013 for example). I am wondering how this may affect the results. I would expect that for most cases, OH and HO2 transferred from gas-phase will be consumed in the surface layer of aerosols or droplets, which is different from the assumption of homogeneous mixing in this work for individual aerosols or droplets.

**Author Response:** The referee is correct that aqueous phase diffusion might impact the overall concentration and the radial concentration of highly reactive compounds such as OH and $HO_2$ - if there were not replenished by additional aqueous phase processes. The phase transfer is not the main source of the radical ROS but their cycling in the aqueous phase (cf Figures 1, 2, and 5 that shows the relative source contributions of chemical reactions and phase transfer, respectively.). The diffuso-reactive parameter $q$ was explored for OH concentrations in cloud and aerosol water using a similar chemcial mechanism as in the present study (Ervens et al., 2014). In this latter study, it was found that $q$ clearly deviated from unity when no aqueous phase sources were considered, with $1.9 < q(OH) < 14.9$ for cloud water, and $32 < q(OH) < 47$ for aerosol water, respectively. However, it was also discussed there that the gradient of OH from the droplet (particles) surface to the center is compensated for by the Fenton reaction if the iron concentration is on the order of $> 10^{-5}$ M in cloud droplets and $> 0.1$ M in aerosol particles. We added the following text to Section 2.1 (Line 95)

*In our model studies, it is assumed that all solutes are homogeneously mixed and no concentration gradients exist between the gas/aqueous interface and the bulk of the aqueous phase. This assumption is justified if chemical loss processes in the aqueous phase are comparatively slow, so that aqueous phase diffusion can evenly distribute the species throughout the aqueous volume. The competition between aqueous phase diffusion and chemical loss in the aqueous phase is commonly quantified by the dimensionless diffuso-reactive parameter q (Equation S.1 in the Supplement, Seinfeld and Pandis (1998)). This parameter does not take into account chemical sources within the aqueous phase. As discussed in a previous study, the Fenton reaction can act as a such a source for the OH radical in aqueous solution (Ervens et al., 2014). A detailed comparison and discussion of the parameter q and the role of aqueous phase diffusion can be found in the Supplement (Section S2).*

Accordingly, we added a new section S2 'Aqueous phase diffusion' in the Supplement:

**S2 Aqueous phase diffusion**

*Gradients between the drop surface and the center of the droplets (or particles) can occur when highly reactive species (e.g. OH, $HO_2$) are taken up from the gas phase into the aqueous phase and are efficiently consumed faster by chemical reactions than aqueous phase diffusion can replenish their concentration throughout the aqueous phase volume. The deviation from a homogeneously mixed aqueous phase is commonly quantified by the diffuso-reactive parameter q (Seinfeld and Pandis, 1998):*

$$q = 0.5\,D \sqrt{\frac{k^{loss}}{D_{aq}}} \tag{S.1}$$

*whereas D is the drop (particle) diameter (20 μm for cloud droplets, 300 nm for aerosol particles in our study), $D_{aq}$ is the aqueous phase diffusion coefficient $2 \cdot 10^2$ $cm^{-2}$ $s^{-1}$ and $k^{loss}$ is the total chemical first-order loss rate in the aqueous phase $(k^{2nd} \cdot [reactant])$ $[s^{-1}]$.*

*The main loss process of OH(aq) is the reaction with WSOC ([WSOC]= $1.3 \cdot 10^{-4}$ M in cloud water, 2.6 M in aerosol wa-*

**Table R-1.** Diffuso-reactive parameter $q$ (Equation S.1) of OH and $HO_2$ radicals in cloud droplets and aerosol particles (pH = 4.5); the concentrations of the reactants of the main loss reactions (WSOC, Fe(III), $HO_2$) are given in M.

| | | Cloud | | Aerosol | |
|---|---|---|---|---|---|
| | OH | [WSOC] | $q$ | [WSOC] | $q$ |
| All droplets (particles) | | $1.3 \cdot 10^{-4}$ | 50 | 2.6 | 105 |
| | $HO_2$ | [Fe(III)] | $q$ | [Fe(III)] | $q$ |
| FeBulk | | $8.2 \cdot 10^{-8}$ | 0.8 | $1.6 \cdot 10^{-3}$ | 1.6 |
| FeN<100, Fe-containing | | $1.7 \cdot 10^{-5}$ | 11 | 0.33 | 24 |
| | | [$HO_2$] | $q$ | [$HO_2$] | $q$ |
| FeN<100, Fe-free | | $2.4 \cdot 10^{-8}$ | 0.24 | $2.2 \cdot 10^{-7}$ | 0.01 |

*ter). We assume that $k^{loss}$ of OH is entirely determined by this reaction (Ervens et al., 2014). The main single loss process of the $HO_2$ radical is the reaction with Fe(III) that contributes to 20% for $F_{N,Fe}$ = 2% and 80% for $F_{N,Fe}$ = 100%. In iron-free droplets and particles, the recombination of $HO_2$ is its main loss process (Section 3.2). Therefore, using the $HO_2$ concentrations as shown in Figures 2 and 3 ($2.4 \cdot 10^{-8}$ M and $2.2 \cdot 10^{-7}$ M) can be assumed, together with an average rate constant of $4.7 \cdot 10^7$ $M^{-1}$ $s^{-1}$ (($k_{HO2+O2-}$+$k_{HO2+HO2}$)/2) is assumed to account for the approximately equal proportions of $HO_2$ and $O_2^-$ at pH = 4.5. Therefore, the first-order loss rates for OH and $HO_2$ can be estimated as:*

$k_{OH}^{loss} = 3.8 \cdot 10^8 \cdot [WSOC]$

$k_{HO2}^{loss} = 1.5 \cdot 10^8 \cdot [Fe(III)]$ *(iron-containing; FeBulk and FeN<100)*

$k_{HO2}^{loss} = 8.3 \cdot 10^5 [HO_2] + 9.7 \cdot 10^7 [O_2^-]$ *(iron-free, FeN<100)*

*The resulting q values are summarized in Table S6. For OH, the diffuso-reactive parameters are 50 and 105 in clouds and aerosol particles, respectively. However, it has been discussed in our previous study that iron concentrations on the order of micromolar (clouds) and molar (aerosol particles) are sufficient to make the Fenton reaction compensating for the rapid OH loss (Table 1 in Ervens (2015)).*

*Generally, the q values of HO$_2$ are smaller than those for OH with a maximum of 24 in iron-containing particles. Similar to the OH formation, also HO$_2$ is efficiently formed in the aqueous phase, with highest rates in iron-containing particles (FeN>100). Therefore, also for the HO$_2$ radical (including its anion), no significant concentration gradients occur between droplet (particle) surface and center. In iron-free droplets and particles, q is less than unity, i.e., even without HO$_2$ formation in the aqueous phase no concentration gradients exist.*

**Referee Comment 1.2:**

While the authors show detailed concentrations of ROS in different scenarios, I am wondering if the authors can also provide gas-phase concentrations of OH, HO$_2$ and H$_2$O$_2$ for each simulation?

**Author Response:** We added the figure below to the the supplement (Figure S3) that shows the gas phase concentrations for OH and HO$_2$ and mixing ratios for H$_2$O$_2$ and added a new Section (3.1.3) to the manuscript:

*3.1.3 Gas phase concentrations (OH, HO$_2$) and mixing ratios (H$_2$O$_2$)*

*The gas phase concentrations of OH and HO$_2$ and mixing ratios of H$_2$O$_2$, are shown in Fig. S3 as a function of time (t $\leq$ 7200 seconds) in the presence of cloud droplets and aerosol particles, respectively, at the three pH values considered here and for the two assumed iron distributions in the aqueous phase. The H$_2$O$_2$ mixing ratio in the presence of cloud water is very quickly reduced to about 50% of its initial concentration (1 ppb) due to partitioning to the aqueous phase. Both the pH value and F$_{N,Fe}$ have a small effect on the H$_2$O$_2$ mixing ratio. The small increase of H$_2$O$_2$ over time is in agreement with findings in a recent multiphase box model intercomparison (Barth et al., 2021).*

*The OH concentrations are on the order of $10^6$ cm$^{-3}$, i.e., typical concentrations as found for continental daytime conditions. This concentrations builds up within a few seconds and does not largely change over the time period considered here. The difference of up to a factor of 5 in the OH concentrations in the presence of cloud droplets with a pH value of 3 and 6, respectively, cannot be directly explained by any OH reactions in the aqueous phase, but are rather a consequence of the OH-HO$_2$ cycle in the multiphase system (Fig. 1). At pH = 6, the effective Henry's law constant for HO$_2$ is highest and therefore, most HO$_2$ is partitioned to the aqueous phase, resulting in the lowest fractions of HO$_2$ in the gas phase. The gas phase HO$_2$ concentrations are on the order of $10^8$ cm$^{-3}$, in agreement with predictions from other atmospheric multiphase chemistry models (e.g., Barth et al. (2021)). Generally, the pH value and iron distributions cause smaller differences in the presence of aerosol particles as compared to the cloud case, since the liquid water content of particles is smaller by several orders of magnitude and therefore a smaller fraction of the species is dissolved.*

**Referee Comment 1.3:** Mass accommodation coefficient and reactive uptake coefficient. The authors show in Table S4 the

[Figure]

**Figure R1-1.** (Figure S3): Gas phase concentrations of (a, d) $H_2O_2$ [ppb], (b,e) OH [molec cm$^{-3}$], (c,f) $HO_2$ [molec cm$^{-3}$]. Upper panels: Cloud simulations; lower panels: Aerosol simulations. The grey vertical line in all panels denotes the time of t = 400 s, at which most of the analyses are performed.

mass accommodation coefficient (alpha) of OH to be 0.05 and $HO_2$ to 0.01. The mass accommodation coefficient is considered to be the upper limit for reactive uptake coefficient. It is surprising that in Figure 8 the derived reactive uptake coefficient for $HO_2$ is higher than the mass accommodation coefficient of $HO_2$ (0.01).

**Author Response:** We thank the referee for noting this inconsistency. We checked our code and realized that an accommodation coefficient of $\alpha_{HO2} = 0.05$ was used. This value is in agreement with the higher values as found in other studies. For example, Morita et al. (2004) have reported a range of $0.01 < \alpha_{HO2} < 0.2$ for the mass accommodation coefficient on water, supported by the experimental studies by Mozurkewich et al. (1987) and Hanson et al. (1992). We added at the end of Section 3.3.1 (Line 440):

*We use identical $\alpha$ values for droplet and aqueous particle surfaces. The mass accommodation coefficient $\alpha$ is the upper limit of the maximum value for the uptake coefficient $\gamma$. This limit can be seen in Fig. 8d where $\gamma_{HO2}$ approaches the value of $\alpha_{HO2}$ = 0.05, i.e., the regime, at which the uptake is fully controlled by the mass accommodation and not by the chemical loss in the aqueous phase. For the $HO_2$ radical, gas phase diffusion may decrease $\gamma_{HO2}$ by a factor of ~2 at pH = 6 whereas the two two $\gamma$ values are nearly identical at pH = 3 (Fig. S8b). In aerosol water, gas phase diffusion has an insignificant impact on its reactive uptake coefficient onto aerosol water (Fig. S8d).*

**Referee Comment 1.4:** Gas-phase diffusion. The equation (E.7) should include a gas-phase diffusion term. Note that for cloud droplets, the reactive uptake process is limited by gas-phase diffusion, not by reactive uptake. Some good explanation on this can be found from Jacob (2000) paper ("Heterogeneous chemistry and tropospheric ozone"). Therefore, the quantification of reactive uptake coefficient (gamma) for cloud droplets could be of minor importance. More importantly, I am wondering if the authors could discuss the importance of gas-phase diffusion throughout this work. Potentially, some of the difference between aerosols and cloud droplets can be attributed to the difference in gas-phase diffusion.

**Author Response:** The referee is correct that gas phase diffusion might be more important for droplets than for aerosol particles, since the characteristic time scales with the square of the droplet or particle radius, respectively. We thank the referee for pointing us to the relevant literature. We replaced the previous Equation E.7 by the extended version E.7rev below, according to Jacob (2000). To demonstrate the effect of gas phase diffusion, we included Figure R1-2 as a new figure in the supporting information (Figure S8).

[Figure]

**Figure R1-2.** (Figure S8): Reactive uptake coefficients for (a, c) OH and (b, d) $HO_2$. Left panels: cloud conditions; right panels: aerosol conditions. The solid lines denote $\gamma$ values based on Equation (7); dashed lines denote $\gamma$ values based on Equation (S.3). The difference between the two values represents the effect of gas phase diffusion.

We moved all discussion of $\gamma$ from Section 3.3.2 to Section 4.2 and changed the text as follows (L. 380ff) (new text in bold):

*The uptake of OH and $HO_2$ into aerosol particles is often parameterized by the dimensionless reactive uptake coefficient $\gamma$*  *that is derived*  **as a function of** *the first-order radical loss ($k^{loss}$) onto aerosol particles.* ***For the derivation of $\gamma$ based on lab studies, it is assumed that $\gamma$ solely depends on the molecular speed and the droplet surface. However, it has been discussed that the reactive uptake coefficient in atmospheric applications should also include a term to account for gas phase diffusion ((Jacob, 2000))***

$$k^{loss} = \gamma \frac{\omega S}{4} \qquad \qquad \text{(7)}$$

$$k^{loss} = \left( \frac{r_d}{D_g} + \frac{4}{\omega \gamma} \right)^{-1} S \qquad \qquad \text{(7rev)}$$

**with S being the total droplet surface area per air volume and $\omega$ the molecular speed,**

$$\omega = \sqrt{\frac{8RT}{\pi M}} \qquad \qquad \text{(8)}$$

*whereas M is the molar mass [g mol$^{-1}$], R is the gas constant, and T is the absolute temperature [K] , and S the aerosol surface, (e.g., Thornton et al., 2005; Pöschl et al., 2007). Accordingly the phase transfer rate of the radicals into the particle phase can be described as*

$$\frac{d[Radical]_g}{dt} = k^{loss}[Radical]_g \qquad \qquad \text{(9)}$$

**and the reactive uptake coefficient $\gamma$ can be derived as**

$$\gamma = \frac{4}{\omega \left( \frac{S}{k^{loss}} - \frac{r_d}{D_g} \right)} \qquad \qquad \text{(10new)}$$

*The $\gamma$ values for OH and HO$_2$ as a function of Fe$_{NFe}$ are displayed in ??. If gas phase diffusion is neglected in the derivation of $\gamma$, the last term in the denominator of Equation S.3 is set to zero (Equation S.3). To demonstrate the impact of gas phase diffusion on the reactive uptake parameter, we compare $\gamma$ values calculated with Equation 10 to values without consideration of gas phase diffusion (Section S3 in the Supplement).*

Using Equation 10new, we re-calculated the uptake coefficients $\gamma$ and replaced Figure 8 accordingly. Since it is actually an interesting finding that gas phase diffusion is much less important for HO$_2$ than for OH, we added the Figure R1-2 in the Supplement and added a new section S2 there:

***S2 Contribution of gas phase diffusion to the reactive uptake parameter $\gamma$***
*For the derivation of $\gamma$ based on lab studies, it is assumed that $\gamma$ solely depends on the molecular speed and the droplet surface (e.g., Pöschl et al. (2007)). Figure R1-2 shows the calculated $\gamma$ values using Equation 7 (identical to Fig. 8 in the main manuscript), together with results using the simpler version of this equation, Equation S.3, that does not include a term to account for gas phase diffusion.*

$$k^{loss} = \gamma \frac{\omega S}{4} \qquad \qquad \text{(S.2)}$$

*and accordingly*

$$\gamma = \frac{S}{k^{loss}} \qquad \qquad \text{(S.3)}$$

*The comparison of the $\gamma$ values demonstrates that the reactive uptake of the OH radical into cloud droplets is significantly controlled by gas phase diffusion; neglecting this effect would lead to an overestimate of the reactive uptake coefficient by a factor of $\sim$3 ($\gamma_{OH} \sim 0.002$ vs $\gamma_{OH} \sim 0.006$). The difference in the $\gamma_{HO2}$ values calculated by Equations 7 and S.3 is negligible for cloud droplets with a pH = 3 whereas it is approximately a factor of 2 at pH = 6. The reason for this trend is the larger $k^{loss}$ rate at higher pH since the $O_2^-$ radical anion reacts faster than the undissociated $HO_2$ radical.*

*The difference in the $\gamma$ values for both radicals is much smaller for aerosol particles, i.e., $\gamma_{OH}$ is overestimated by $\sim$10% if gas phase diffusion is neglected (0.038 vs 0.045 in Figure S8c). The values for the $HO_2$ radical, $\gamma_{HO2}$, are nearly indistinguishable using both equations (Equation 7 and S.3, Figure S8d). Also under aerosol conditions, the pH value and $F_{N,Fe}$ have a much greater influence on $\gamma_{HO2}$ than gas phase diffusion. This analysis demonstrates that gas phase diffusion should be taken into account for the reactive uptake of the radical into cloud droplets as suggested previously ((Jacob, 2000)), whereas the role for uptake onto aerosol particles is of minor importance.*

**Technical Comments:**

**Referee Comment 1.5:**Line 69 "…perform box model simulations a box model with…", remove "a box model"

**Author Response:** The sentence was corrected: 'a box model' was removed.

**Referee Comment 1.6:**Line 90 Equation 1-3 needs a reference. How are they derived?

**Author Response:** Equations 1-3 are based on the equations, for which detailed derivations are given in atmospheric chemistry textbooks )e.g., Seinfeld and Pandis (1998); Eqs 11.116, 11.122, 11.123). In our model, however, we use consistently units of $mol\,g_{air}^{-1}$ for all species concentrations in both phases. The box model used for this study is part of a larger cloud microphysics/chemistry parcel model that describes the trajectory of an air parcel in cloudy and cloud-free air, e.g., (Feingold and Heymsfield, 1992; Ervens et al., 2003). As the air density changes during lifting and descent of an air parcel, concentrations related to air volume (e.g., molec cm$^{-3}$) are not practical, since the air density (and with it volume) changes with temperature and pressure changes. Therefore, units of $[mol g_{air}^{-1}]$ are applied that are independent of air volume changes.

In the current box model application, they can be simply converted using the constant air density of 0.0013 g cm$^{-3}$.

For clarity, we show below the conversion of the equations as given by (Seinfeld and Pandis, 1998) ('SP') to those as used in our model. Because these are just unit conversions using standard equations (e.g., law for ideal gases etc.), we did not include the derivation in the manuscript.

The mass transfer coefficient $k_{mt}$ takes into account gas phase diffusion and the mass accommodation, i.e. the probability that a molecule that hits the droplet (or particle) surface is taken up. We added the description of the two terms as follows after Equation 3 in the manuscript at L. 92ff (new text in bold):

*The mass transfer coefficient $k_{mt}$ takes into account gas phase diffusion and the mass accommodation, i.e. the probability that a molecule that hits the droplet (or particle) surface is taken up.* **It is the inverse of the characteristic time for these two resistances. Its full derivation can be found in the literature, e.g. by (Nathanson et al., 1996), and is therefore not repeated here.**

**Derivation of model equations**

In the textbook by Seinfeld and Pandis (1998), the following equation is given:

$$\frac{dp}{dt} = -k_{mt}\,LWC\,p + \frac{1}{K_{H(eff)}}\,k_{mt}C_{aq}\,LWC \tag{E.11.122SP}$$

whereas the gas phase partial pressure $p$ in *atm*, $C_{aq}$ is the aqueous phase concentration in mol $L_{aq}^{-1}$, and *LWC* denotes the liquid water volume fraction [vol(aq)/vol(air)]. According to the law for ideal gases

$$p = C_g\,\rho_{air}\,RT\,1000 \tag{E.conv1}$$

with $p$ in [*atm*], $C_g$ in [$mol\ g_{air}^{-1}$], $\rho_{air}$ in [$g\ cm^{-3}$], $R = 0.082\ L\ atm\ /$ and $T$ in $K$; the factor 1000 accounts for the conversion from cm$^{-3}$ to L.

Replacing further the aqueous phase concentration $C_{aq}$ [$mol\ L_{aq}^{-1}$] by the equivalent in terms of gas phase concentration $C_{aq}^g$ [$mol\ g_{air}^{-1}$]

$$C_{aq} = \frac{C_{aq}^g \rho_{air}\,1000}{LWC} \tag{E.conv2}$$

Including E.conv1 and E.conv2 in E.11.122SP' yields

$$\frac{dC_g}{dt}\rho_{air}RT\,1000 = -k_{mt}\,LWC\,C_g\rho_{air}RT\,1000 + \frac{1}{K_{H(eff)}}\,k_{mt}\,LWC\frac{C_{aq}^g\rho_{air}\,1000}{LWC} \tag{E.2'}$$

Dividing both sides of the equation by a factor ($\rho_{air}/(RT1000)$ and rearranging the terms in E.2', one obtains

$$\frac{dC_g}{dt} = -k_{mt}\,LWC_m\left(C_g - \frac{C_{aq}^g}{LWC\,K_{H(eff)}RT}\right) \tag{E.2''}$$

Since the concentration change of a gas phase species is not only due to the transfer as described by Eq.2'' but also due to chemical reactions, the complete model equation is

$$\frac{dC_g}{dt} = -k_{mt}\,LWC\left(C_g - \frac{C_{aq}^g}{LWC\,K_{H(eff)}\,RT}\right) + S_g - L_g \tag{E.2}$$

whereas $S_g$ and $L_g$ are the chemical source and loss rates of the chemical species.

The phase transfer of an aqueous phase species is described by Seinfeld and Pandis (1998) as

$$\frac{dC_{aq}}{dt} = \frac{k_{mt}}{RT}p - \frac{k_{mt}}{K_{H(eff)}RT}C_{aq} - Loss_{aq} \tag{E.1.123SP}$$

E.1.123SP can be converted correspondingly to obtain Eq.1 as used in the code.

$$\frac{dc_{aq}}{dt} = k_{mt}\,LWC\left(C_g - \frac{c_{aq}}{LWC\,K_{H(eff)}\,RT}\right) \tag{E.1}$$

**Referee Comment 1.7:** Line 108 why 400 seconds?

**Author Response:** We chose a simulation time of 400 seconds as this roughly corresponds to the lifetime of a single cloud droplet. This time can be estimated based on updraft and downdraft velocities a cloud droplet experiences and on cloud thickness. For example, in a cloud that is 100 m thick, with average vertical velocity of 0.5 m s $^{-1}$, the lifetime of a droplet is 400 s (2 x 100 m x 0.5 m s $^{-1}$).

To demonstrate that our results and conclusions do not largely depend on the choice of the timescale, we compare the results of Figure 5 and S5 whereas the latter figure was created using results at t = 2000 s. Also the new Figure SX (gas phase concentrations) in the supporting information shows that the concentrations do not largely vary between 400 and 2000 s or on even longer time scales (t $\leq$ 7200 s). It should be noted though that we do not consider any replenishment of initial gases. This may not be realistic for longer time scales since highly reactive gases are quickly depleted (e.g., $SO_2$) whereas in the real atmosphere, emissions might replenish them.

However, for the conceptual model studies presented here, the assumption of initial concentrations (Table S5) seems sufficient as it has also been done in previous box model studies (e.g.Barth et al. (2021)).

We added the following text to the manuscript to further clarify the choice of 400 s (Section 2.2), Line 118:

*Chemical and phase transfer **rates** are analysed after a simulation **time** of 400 seconds. **This time scale refers approximately to the lifetime of a single cloud droplet. It can be estimated based on updraft and downdraft velocities a cloud droplet experiences and on cloud thickness (e.g., average vertical velocity of 0.5 m s$^{-1}$ throughout cloud that is 100 m thick, as typical values for shallow stratocumulus clouds).** Gas phase concentrations **are initialized with the values in Table S6 and** are not replenished throughout the simulation. **The three ROS do not show a strong dependence on time (Fig. S3; Section 3.1.3). To** demonstrate that our results and conclusions do not strongly depend on the choice of the timescale, we compare the  **main source and loss processes of the three species** at t = 400 s and t = 2000 s (Section 3.2) **as** also the concentrations of OH, $HO_2$ and $H_2O_2$ do not differ largely over this time scale.  During even longer time scales, the concentrations may change in the box model; however, this may not be realistic  since highly reactive gases are quickly depleted (e.g., $SO_2$) whereas in the real atmosphere, emissions might replenish them. *

**Referee Comment 1.8:** Line 125 "Fig. 1" should be "Fig. 2".
**Author Response:** The referee is right. We corrected it.

**Referee Comment 1.9:** Table 1 Iron-containing aerosol particles should be "30", not "2".
**Author Response:** We thank the referee for spotting this typo. We corrected it so that the total aerosol particle number concentrations correctly adds up to 1500 $cm^{-3}$.

**Referee Comment 1.10:** Line 212-214, this should go to Line 108. "This time corresponds approximately to the lifetime of a single cloud droplet but underestimates the time an aerosol particle might be exposed to a given relative humidity. We have chosen this relatively short time, as in our box model setup, the initialized gases are not replenished over time as no emissions are considered."

**Author Response:** We agree with the referee that the text was distracting here. We removed the corresponding text but clarify earlier (see our response to Comment 1.7 above).

**Referee Comment 1.11:** Line 387: It seems that Figure 6 should go after Figure 8.

**Author Response:** We agree with the referee that the discussion of Figure 6 is much better placed in Section 4.1. Therefore, we moved it there and changed the figure numbering accordingly and shortened the text at the end of Section 3.2:

Line 264: *Therefore, only a difference in the $Fe^{2+}$ concentration can explain the differences in the chemical rates of the chemical $H_2O_2$ sources and sinks (**Section 4.1**). Indeed, Fe(II) contributes only ~20% to the total iron for $F_{N,Fe}$ = 2% whereas its contribution is ~55% for the FeBulk approach ($F_{N,Fe}$ = 100%) (Fig. X*

Line 286: *The highly efficient loss of $O_2^-$ at high pH results in the lowest $HO_2/O_2^-$ concentrations in the iron-containing droplets at pH = 6 (Fig. S2d). Since the reactions of Fe(III) with $HO_2/O_2^-$ are the main reduction processes of Fe(III), this low $HO_2/O_2^-$ concentration leads to inefficient Fe(III) to Fe(II) conversion and relatively higher Fe(III) concentrations (**Section X**). This is reflected in the lower predicted Fe(II)/Fe(total) ratio at low $F_{N,Fe}$ as compared to the FeBulk approach (Fig. XX). The difference in this ratio between $F_{N,Fe}$ = 2% and 100% is about a factor of 2 for cloud water and up to a factor of 3 in aerosol particles. Consequences of this finding for model and field studies are discussed in Section 4.1.*

**Comments by Referee 2**

In the manuscript at hand, Khaled and co-workers investigate the effect of iron concentration heterogeneity within a population of aerosol or cloud particles on the overall concentrations and turnover rates of reactive oxygen species (ROS) using a kinetic model. This idea is novel and unique and the results are quite significant. The topic fits well within the scope of ACP and should be interesting to most readers. The biggest challenge of this work is to communicate the model setup and results clearly, and to derive general conclusions. The paper is written well from a technical perspective, but is a slow and cumbersome read due to the inherent complexity of the model calculations and some inaccuracies in language and writing. I can recommend publication of this article after a few points, outlined below, are addressed.

**Author Response:** We thank the referee for their positive evaluation and constructive comments on our manuscript. We address all referee comments point-by-point below. Line numbers refer to the new manuscript version without track-change. In addition, we improved the text and structure throughout the manuscript; these changes are reflected in the track-change version of the revised manuscript and not listed in detail here.

**Referee Comment 2.1:** Role of the gas phase: The gas phase is a source of ROS in most of the presented calculations, but l. 108 states that gas phase concentrations are not replenished. Can the authors comment on the initial conditions of ROS in the gas phase, what were they based on? What is the time evolution / how strongly are gas phase ROS concentrations affected by presence of cloud and aerosol particles? Is a steady state achieved? How different is this steady state from the initial conditions?

**Author Response:** Since also Referee 1 asked for about the gas phase concentration, we added a new figure in the supplement (Figure S3 in the supporting information) showing gas phase concentration of the three key species (OH, $HO_2$, $H_2O_2$). In the caption of Figure S3, we point out that radicals are not initialized but only stable trace gases, as listed in Table S6. Such initialization is common for photochemical models and has been applied in many previous box models studies, including a recent box model intercomparison (e.g., Ervens et al. (2014); Barth et al. (2021). To describe the new Figure S6, we added a new Section (3.1.3) to the manuscript:.

*3.1.3 Gas phase concentrations (OH, $HO_2$) and mixing ratios ($H_2O_2$)*

*The gas phase concentrations of OH and $HO_2$ and mixing ratios of $H_2O_2$, are shown in Fig. S3 as a function of time ($t \leq 7200$ seconds) in the presence of cloud droplets and aerosol particles, respectively, at the three pH values considered here and for the two assumed iron distributions in the aqueous phase. The $H_2O_2$ mixing ratio in the presence of cloud water is very quickly reduced to about 50% of its initial concentration (1 ppb) due to partitioning to the aqueous phase. Both the pH value and $F_{N,Fe}$ have a small effect on the $H_2O_2$ mixing ratio. The small increase of $H_2O_2$ over time is in agreement with findings in a recent multiphase box model intercomparison (Barth et al., 2021).*

*The OH concentrations are on the order of $10^6$ $cm^{-3}$, i.e., typical concentrations as found for continental daytime conditions. This concentrations builds up within a few seconds and does not largely change over the time period considered here. The difference of up to a factor of 5 in the OH concentrations in the presence of cloud droplets with a pH value of 3 and 6, respectively, cannot be directly explained by any OH reactions in the aqueous phase, but are rather a consequence of the OH-$HO_2$ cycle in the multiphase system (Fig. 1). At pH = 6, the effective Henry's law constant for $HO_2$ is highest and therefore, most $HO_2$ is*

[Figure]

**Figure R2-1.** (Figure S3): Gas phase concentrations of (a, d) $H_2O_2$ [ppb], (b,e) OH [molec cm$^{-3}$], (c,f) $HO_2$ [molec cm$^{-3}$]. Upper panels: Cloud simulations; lower panels: Aerosol simulations. The grey vertical line in all panels denotes the time of t = 400 s, at which most of the analyses are performed.

*partitioned to the aqueous phase, resulting in the lowest fractions of $HO_2$ in the gas phase. The gas phase $HO_2$ concentrations are on the order of $10^8$ cm$^{-3}$, in agreement with predictions from other atmospheric multiphase chemistry models (e.g., Barth et al. (2021)). Generally, the pH value and iron distributions cause smaller differences in the presence of aerosol particles as compared to the cloud case, since the liquid water content of particles is smaller by several orders of magnitude and therefore a smaller fraction of the species is dissolved.*

**Referee Comment 2.2:** L. 547 states "We find that $H_2O_2$ concentrations in cloud water are not affected by FN,Fe as $H_2O_2$ is in thermodynamic equilibrium between the gas phase and all droplets". Is this because the gas phase is a (near-)infinite source of H2O2?

**Author Response:** The loss of hydrogen peroxide in the aqueous phase is not as efficient as for the highly reactive radicals (OH, $HO_2$). Therefore, the transport of $H_2O_2$ into the aqueous phase is sufficiently fast to replenish the reacted $H_2O_2$ there. In addition, the formation of $H_2O_2$ in the aqueous phase is very efficient, even exceeding the source by phase transfer (Figures 2, 3, and 5a). The main source and loss reactions of $H_2O_2$ are independent of iron, i.e. reaction S1 (recombination of $HO_2$) and L1 (S(IV) oxidation), respectively (Figure 5a). Therefore, the iron distribution or iron concentration does not strongly effect the $H_2O_2$ levels. To clarify this, we added in line 545:

*We find that $H_2O_2$ concentrations in cloud water are not affected by $F_{N,Fe}$. **Its main source and loss processes include the recombination of $HO_2$ and the reaction with S(IV), respectively. Since the loss process is slower than the uptake from the gas phase and the production in the aqueous phase,**  $H_2O_2$ is in thermodynamic equilibrium between the gas phase and all droplets.*

**Referee Comment 2.3:** What I could not understand when reading this paper: Why are the OH concentrations in cloud droplets lower for the FeBulk case, compared to both, the iron-rich and the iron deprived particles in the FeN<100 case? (l. 144-149).

**Author Response:** To understand the differences in the OH concentrations between (i) iron-containing and iron-free droplets and (ii) FeBulk and FeN<100, we refer to Figure 5b. The higher OH concentration in the two different droplet classes in the FeN<a00 approach as compared to FeBulk has to separate reasons:

1) In the *iron-free droplets*, the main source of the OH radical is the reaction of ozone with $O_2^-$ (S1). The loss of $HO_2/O_2^-$ by recombination (L1, Figure 5c) is rather inefficient. Therefore, the $HO_2/O_2^-$ concentration in iron-free droplets is higher than in iron-containing droplets - resulting in efficient OH formation.

2) In the *iron-containing droplets*, the iron concentration in the FeN<100 approach is much higher than in the droplets in the FeBulk approach: The total iron concentration scales inversely with the LWC (i.e. it is 49 times higher in the 2% of the droplets); however, since the fraction of Fe(II) to total iron is about a factor of 2 lower, one can conclude that the Fe(II) concentration in the iron-containing droplets in the FeN<100 approach is about 25 times higher than in the FeBulk approach. Therefore, the rate of the Fenton reaction (S2, Figure 5b) is higher in the FeN<100 approach, leading to more OH in the iron-containing droplets than in the bulk approach.

We added to the text (L. 153):

*The reasons for these differences are further explored in Section 3.2 where the individual chemical processes affecting the concentrations are analyzed.*

**Referee Comment 2.4:** The results of the model simulations are highly complex and I wonder if they could be reduced / simplified in the main text of this manuscript. Is the presentation of two iron concentrations mFe = 10 and 50 ng m-3 in the main text really necessary? This addition makes Figs. 6 and 7 very dense.

**Author Response:** We agree with the referee that Figure 6 and 7 contained a lot of information. We removed the lines for $m_{Fe}$ = 50 ng $m^{-3}$ from the figures. However, we still discuss briefly these results because we think that it is an important finding and compare the presults for the Fe(II)/Fe(total) ratio and bulk aqueous phase concentrations for both $m_{Fe}$ in Figures S5 and S7 in the Supplement. It seems an important finding that our results are largely independent of the total iron mass.

We added in the manuscript (Line 303):

*This trend can be explained because (i) in cloud water the main chemical source and loss processes are independent of iron, whereas (ii) in aerosol water, both the main chemical source and loss reactions are dependent on the Fe(II) concentration. The same set of processes can also explain why the resulting aqueous phase $H_2O_2$ concentrations are independent of the total iron mass. In a sensitivity study, we increased $m_{Fe}$ to 50 ng $m^{-3}$ which results in very similar $H_2O_2$ concentrations (Fig. S5 and S7).*

**Referee Comment 2.5:** What does the presentation of epsilon values for many different conditions (Fig. 3) really add to the paper?

**Author Response:** The partitioning coefficient $\epsilon$ is a key parameter that can explain several of the findings in our study. It is a transition from the description of the concentrations in Figures 2 and 3 and the detailed analysis of the individual sources and loss processes in Figure 5. To highlight the importance of this parameter in our discussion, we shortened Sections 3.1.1. and 3.1.2 and discuss in a new Section 3.1.4 the partitioning coefficient $\epsilon$ (see also our response to Comment 2.16).

Main conclusions based on this figure include:

1) The fact that $\epsilon_{H2O2}$ is $\sim$1, shows that $H_2O_2$ is in thermodynamic equilibrium and explains why the $H_2O_2$ concentrations do not differ between iron-containing and iron-free droplets (particles).

2) The $\epsilon_{OH}$ values for iron-containing aerosol particles show a distinct trend with highest $\epsilon_{OH}$ values in iron-containing particles (FeN<100, FeBulk) whereas $\epsilon_{OH}$ in iron-free particles is lowest. This trend implies that an iron reaction leads to an efficient formation of OH, i.e. compensating its loss in the aqueous phase and/or its inefficient uptake. The differences in $\epsilon_{OH}$ scale with the OH concentrations (Figure 4) that differ by three orders of magnitude between iron-containing and iron-free particles in the FeN<100 approach and the concentration in the FeBulk case. The $\epsilon_{OH}$ values in cloud water are much closer together which is also reflected in the more similar concentrations in all droplets (Figure 4).

3) The lack of a clear trend in $\epsilon_{OH}$ with pH suggests that the main formation and loss processes of OH are pH independent.

4) For $\epsilon_{HO2}$, iron reactions cause a significant deviation from thermodynamic equilibrium at high pH whereas the $HO_2$ partitioning shows a decreasing trend with increasing pH. The decreasing $\epsilon_{HO2}$ with increasing pH suggests that $HO_2$ is more efficiently consumed at high pH - which is in agreement with the general trend that reactions with the $O_2^-$ radical anion are faster than those with the $HO_2$ radical.

5) Unlike for OH, for which the smallest $\epsilon$ is seen in iron-free particles, the lowest $\epsilon_{HO2}$ is found for iron-containing particles - which shows that the concentration differences due to iron distribution for OH and $HO_2$ have different reasons: Whereas iron leads to efficient OH formation, it causes significant $HO_2/O_2^-$ loss.

**Referee Comment 2.6:** I worry about the conclusions being made on oxidative potential and health effects of particles. ROS species are generally not thought of as being inhaled with the particles, but rather being generated in situ upon inhalation of redox-active PM constituents (Lakey et al. 2016). This should be especially true for transient species such as OH and $HO_2$. For H2O2, it would be interesting to see an estimate: How does the amount of H2O2 that is present in these particles compare to the H2O2 production that occurs after dissolution of PM in the lung (e.g. Tong et al. 2016)?

**Author Response:** The referee is right that the total ROS concentration of inhaled particles is likely of less concern regarding particles' health effects than the production during their presence in lung fluid.

We agree that a comparison of $H_2O_2$ production while particles are dissolved in lung fluid vs in air might be interesting. However, since these are two very different regimes (i.e. lung fluid as a bulk system vs multiphase (gas/aqueous) system for particles in the atmosphere), such an estimate is not trivial and straightforward. We predict a net uptake of $H_2O_2$ from the gas phase into droplets and particles for nearly all cases (Figures 2, 4, S1, S2) consider a very different system here than it would be for a more closed system as represented by lung fluid. To address the referee's concern regarding the particle dissolution upon inhaltation, we shortened the text as follows (Line 514): *Our model studies suggest that the total iron amount within*

*an aerosol population may not be a sufficient parameter to constrain the oxidant concentrations within particulate matter **in the atmosphere. However, the resulting health impacts of inhaled particles will likely not be largely affected by $F_{N,Fe}$ since upon inhalation, iron will likely dissolve in the lung fluid so that the particle-resolved effects of the iron distribution are diminished. While it may be possible that there are concentration gradients within the lung fluid due to dissolution and/or mixing kinetics, such effects cannot be quantified to date.***. $H_2O_2$ is often considered one of the main contributors to the total oxidant budget in particles (Tong et al., 2016). Given the large uncertainty in the partitioning of $H_2O_2$ into aerosol water (Section XXX), the extent is not clear to which the iron distribution affects its total concentration in the particles phase. The $H_2O_2$ budget may be overestimated by up to an order of magnitude if the enhanced Henry's law constant ($K_{Heff,H2O2} = 2.7 \cdot 10^8$ M atm$^{-1}$) is more appropriate (Fig. S4). We find that the OH concentration in particles can be over- or underestimated at low $F_{N,Fe}$, depending on $K_{Heff,H2O2}$ (Fig. S4b), whereas the $HO_2$ may be overestimated if $K_{Heff,H2O2}$ is higher than the physical Henry's law constant (Fig. S4c). The partitioning of $H_2O_2$ into aerosol water (i.e. $K_{Heff,H2O2}$) is likely dependent on the aerosol composition. Therefore, this dependence together with the [ROS] dependence on $F_{N,Fe}$ might explain the different conclusions regarding the trends of ROS budgets in particles of different air masses. Similar to the Fe(II)/Fe(III) ratio, our model results suggest that bulk samples of aerosol particles may not accurately represent the ROS budget in ambient populations. Upon sampling, the particles might get dissolved and undergo the ROS formation and conversion processes. Several studies have suggested efficient ROS formation from secondary organic material (e.g., Tong et al., 2018; Zhou et al., 2019; Wei et al., 2021). The underlying formation processes are not fully clear; given that SOA is likely present in all particles (and droplets), the effect of iron distribution on ROS formation might become less important if ROS yields from purely organic particles are sufficiently high. However, it has been shown that ROS formation by SOA can be enhanced in the presence of metal ions (Nguyen et al., 2013; Tuet et al., 2017). While the ROS budgets of atmospheric particulate matter might be different depending on these parameters, these uncertainties might be diminished upon inhalation: Firstly, all iron will be dissolved such that the solution can be described by FeBulk. Secondly, the reaction medium is limited to a bulk liquid phase (lung fluid) and thus $H_2O_2$ gas/aqueous partitioning does not play a role.*

**Referee Comment 2.7:** In line 510, I do not understand the meaning of this sentence: "Therefore, this dependence together with the [ROS] dependence on FN,Fe might explain the different conclusions regarding the trends of ROS budgets in particles of different air masses."

**Author Response:** We agree with the referee that this sentence was somewhat convoluted. Note that the full paragraph was removed during the revision.

**Referee Comment 2.8:** l. 549 reads: "However, since H2O2 has been found to partitioning more efficiently into aerosol than into pure water (KHeff,H2O2 = 2.7·108 M atm-1), We find that H2O2 concentrations in cloud water might be underestimated by the FeBulk approach whereas it maybe overestimated in aerosol water." I do not understand this sentence (and there are several typos; "partitioning" -> partition, "We" -> we, "it" -> they, and "maybe" -> may be). How can cloud water be affected by a parameterization that is used for aerosol? Prior it was stated that "H2O2 concentrations in cloud water are not affected by FN,Fe" (l. 547).

**Author Response:** We agree with the referee that this sentence was misleading. We simplified it as follows (L. 549):

*However, since $H_2O_2$ has been found to partitio more efficiently into aerosol **water** than into pure water  **applying a higher effective Henry's law constant ($K_{Heff,H2O2}$ = 2.7·10$^8$ M atm$^{-1}$), results in an overestimate of the** $H_2O_2$ concentrations  **in aerosol** water  by the FeBulk approach *

**Technical Comments**

**Referee Comment 2.9:** There are several inconsistencies with italicized vs. non-italicized variables and constants. **Author Response:** We carefully checked the full manuscript and corrected the formatting where necessary. We do not list all instances here, but they are reflected in the 'track change' file.

**Referee Comment 2.10:** l. 12 – there is a superfluous "of" in "As the main reduction of process of Fe(III)"
**Author Response:** The referee is correct. We removed the 'of'

**Referee Comment 2.11:** l. 69 – "We perform box model simulations a box model with a detailed gas and aqueous phase chemical mechanism" – sentence seems missing a word.
**Author Response:** We removed 'a box model' to correct the sentence.

**Referee Comment 2.12:** l. 88 – Loss rates in units gair-1 is a bit curious. What is the advantage here, would it not better to use air volume?
**Author Response:** The box model as applied in the current study is part of an parcel model to simulate detailed cloud microphysics and chemistry in an air parcel that enters and leaves a cloud (e.g. Ervens et al., 2004). Along the parcel's trajectory, air density changes due to temperature and pressure changes. Therefore, in the original model all species concentrations are given in $mol\,g_{air}^{-1}$ as this unit is conservative, i.e. concentrations do not change with air density. In the current box model set-up in which temperature and pressure do not change during the simulations, the concentrations can converted using the air density (0.00131 $g\,cm^{-3}$ at T = 281.9 K and p = $9.21 \cdot 10^4$ Pa. We added this information in Section 2.1., L. 104:
*The simulations are run at constant temperature (285.6 K), pressure (9.21·10$^4$ Pa) and air density (1.31·10$^{-3}$ g cm$^{-3}$)*

**Referee Comment 2.13:** l. 126 – There seems to be a word missing, please explain - "For all three ROS, the ratio of the phase transfer rates near the ratio of the LWCs of the two droplet classes (98% : 2%)."
**Author Response:** The referee is correct. The verb was missing. We corrected the sentence (L. 139):
*For all three ROS, the ratio of the phase transfer rates **nearly corresponds to**  the ratio of the LWCs of the two droplet classes (98% : 2%).*

**Referee Comment 2.14:** Figure 2: The boxes at the bottom after "bulk aqueous phase concentrations" confused me at first as they appear to be part of the figure legend below. A stronger visual divide between the two might be worthwhile.
**Author Response:** We thank the referee for this suggestion. We re-ordered the elements in Figure 2 (and correspondingly Figures 3, S1 and S2), such that it is clear the the bulk aqueous phase concentrations only refer to panels b (and to b and d in

the supplemental figures) whereas the legend refers to the full figures. As an example, we added the revised Figure 2 below; all other figures can be found in the revised manuscript and supporting information.

[Figure]

**Figure R2-2.** (Figure 2:)Multiphase scheme for cloud water showing the phase transfer and chemical source and loss rates in the aqueous phase, $m_{Fe}$ = 10 ng m$^{-3}$, pH = 4.5 a) FeBulk approach, b) FeN<100 approach with a fraction of iron-containing droplets of $F_{N,Fe}$ = 2%. Numbers below the species names are aqueous phase concentrations [M], chemical and phase transfer rates are shown in M s$^{-1}$. Net phase transfer rates near the top of the figure are expressed in gas phase units [mol $g_{air}^{-1}$ s$^{-1}$]; the contributions [%] into the iron-free and iron-containing droplets are shown next to the arrows. The bulk concentrations below panel b) represent the total ROS concentrations weighted by the contributions of the two droplet classes to total LWC (98% : 2%).

**Referee Comment 2.15:** In l. 150, bulk aqueous phase concentrations are referred to as "at the bottom of Fig. 2b", but that is where the legend is.

**Author Response:** We hope that the revised Figures 2, 3, S1, and S2 resolve this issue. In addition, we changed the text in all figure captions

**Referee Comment 2.16:** Figure 3 is quite confusing and raises more questions than it answers. After several minutes looking at this figure, I am not sure what I learned. Is epsilon a useful parameter or would total concentrations be easier to understand and put in context?

**Author Response:** We thank the referee for this suggestion. However, we think that the total concentration would not convey the same message as showing the partitioning coefficient. The partitioning coefficient $\epsilon$ is independent of total concentrations, but it is a measure of the deviation from thermodynamic equilibrium, i.e., if $C_{aq}$ and $p_g$ in Equation E.4 were equilibrium concentrations according to Henry's law, $\epsilon = 1$. This parameter has often been used in atmospheric chemistry model studies to explore whether individual compounds are in equilibrium (e.g. Ervens (2015); Barth et al. (2021)).

The increasing deviation from $\epsilon_{HO2} = 1$ in Figure 4c with increasing pH in iron-containing droplets (particles) is a key result of our study that explains several other trends.

To emphasize the importance of this parameter for the discussion of our results, we restructured Section 3.1 and discuss now

the partitioning coefficient in a separate Section 3.1.4 'Partitioning coefficient $\epsilon$' moving some of the text of the previous Sections 3.1.1 and 3.1.2 there and adding some more detail. In addition, we changed the colors in Figure 4 since we realized that the referee might have been misled due to identical colors (blue, red) that had a different meaning here (cloud vs aerosol) as opposed to all all other figures where red and blue indicated different pH values.

[revised manuscript text omitted]